# Tug-of-War No More: Harmonizing Accuracy and Robustness in Vision-Language Models via Stability-Aware Task Vector Merging

**Junhao Dong[1,2], Xinghua Qu[3], Cong Zhang[3], Qi Rong Sua[1], Nguyen Duc Thai[1]**
**Wenbo Pan[4], Xinfeng Li[1*], Tongliang Liu[5], Piotr Koniusz[6,7], Yew-Soon Ong[1,2*]**
[1]Nanyang Technological University, [2]Centre for Frontier AI Research, IHPC, A*STAR,
[3]Bytedance,[4]City University of Hong Kong, [5]University of Sydney,
[6]University of New South Wales, [7]Data61♥CSIRO
junhao003@ntu.edu.sg

## Abstract

Foundation Vision-Language Models (VLMs) excel across benchmarks yet remain vulnerable to adversarial attacks. While adversarial fine-tuning improves robustness, attaining a desirable clean–robust performance trade-off typically requires costly hyperparameter searches with multiple retraining runs. A promising alternative is to merge task vectors (*i.e.*, parameter displacements from pretrained models) to balance accuracy and robustness without retraining. However, we find that naive task-vector merging produces a near-linear trade-off, as it equally weights all coordinates and fails to distinguish weights that aid both objectives from those that create conflicts. To overcome this limitation, we propose a prediction stability-aware merging framework that composes task vectors from off-the-shelf naturally and robustly fine-tuned VLMs. Our key insight is that *prediction stability* serves as a proxy for cross-objective compatibility, enabling us to favor perturbation-invariant parameters while attenuating those with high cross-objective impact. Specifically, we estimate per-parameter stability from gradients under both objectives, building complementary masks that retain jointly stable coordinates while suppressing counterpart-sensitive ones. We further refine these masks along adversarial parameter trajectories, with steps weighted by a prediction-sensitivity index. Our theoretical analysis shows that the masks provably contract first-order cross-objective interference, and the prediction criticality index tracks curvature, biasing the merge toward flatter minima and better generalization. Extensive experiments across benchmarks and scenarios demonstrate our method consistently achieves superior clean–robust trade-offs over prior approaches, with the learned balance transferring effectively to downstream tasks.

## 1 Introduction

Despite redefining multimodal learning across diverse tasks, foundation Vision-Language Models (VLMs) like CLIP (Radford et al., 2021) remain alarmingly vulnerable under adversarial attacks (Zhang et al., 2022; Zhao et al., 2023). Even subtle input perturbations can trigger huge performance drops, undermining their reliability in practice and posing severe security risks (Huang et al., 2025).

Bridging the gap between natural performance and robustness is thus essential for the safe and widespread deployment of VLMs. Previous efforts primarily focused on adversarial fine-tuning, where adversarial examples are adaptively integrated into training to enhance robustness (Mao et al., 2023; Schlarmann et al., 2024; Dong et al., 2025b;c;e). However, extensive empirical evidence indicates that even increasingly larger and advanced multimodal architectures continue to suffer from a persistent accuracy-robustness trade-off (Wang et al., 2024). Rather than resolving this fundamental tension, most existing approaches rely on exhaustive hyperparameter searches and costly retraining to find acceptable compromises, limiting the scalability and efficiency of robust VLM solutions.

---

*Corresponding authors: Xinfeng Li (lxfmakeit@gmail.com), Yew-Soon Ong (asysong@ntu.edu.sg).

Given recent progress in parameter-space model merging, which combines fine-tuned models without joint training (Wortsman et al., 2022b; Ilharco et al., 2023), a compelling question arises: *Can model merging extend beyond similar tasks to reconcile the inherent conflict between natural performance and adversarial robustness?* However, our initial investigation reveals that vanilla task-vector merging of the vision encoder in VLMs yields a near-linear clean–robust trade-off with no sweet point, as empirically shown in Section 3.2. To gauge feasibility and diagnose the issue, we examine directional compatibility by comparing gradients of the two objectives at the respective fine-tuned CLIP models. Figure 1 reports *gradient sign agreement* and *gradient cosine similarity* between natural and adversarial losses, where higher values indicate a more similar update direction to preserve clean accuracy while improving robustness. To test whether any observed alignment is merely local, we evaluate both **local gradients** (at the fixed parameters) and **neighborhood gradients** averaged within an $\ell_2$-ball of radius $\epsilon$, thereby probing stability under small parameter changes. Although this analysis reveals a degree of alignment, it also shows that the alignment remains modest and degrades with a larger attack radius, evidencing growing directional conflict and motivating a more selective, stability-aware merging strategy over naive uniform addition.

Motivated by the need to resolve the parameter-level conflicts, we, for the first time, propose a novel model merging framework based on task vectors (*i.e.*, parameter differences between fine-tuned and pre-trained models) derived from *off-the-shelf* naturally and adversarially fine-tuned models, named **PredIction STability-aware mOdeL mErging (PISTOLE)**, to reconcile natural performance and robustness without repeated fine-tuning by selectively fusing compatible knowledge. Specifically, our PISTOLE estimates per-parameter stability under the natural and robust objectives from gradient magnitudes and builds complementary, gradient-informed masks that retain coordinates stable for both objectives while down-weighting those that the counterpart would strongly change. These masks are applied to the respective task vectors prior to mixing.

To better capture local loss-parameter geometry, we refine the masks by accumulating gradients along adversarial parameter trajectories, with steps weighted by a prediction-sensitivity index that quantifies how even small parameter perturbations affect the output. Furthermore, we provide theoretical analyses demonstrating that these masks contract cross-objective first-order interference and that the sensitivity index tracks curvature, steering the merge toward flatter, more generalizable regions and yielding a stronger clean–robust trade-off.

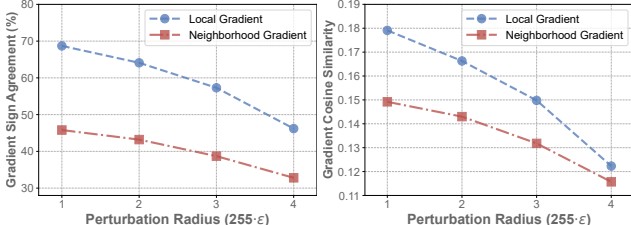

(a) Gradent Sign Agreement    (b) Gradient Cosine Similarity

Figure 1: Gradient alignment on ImageNet between naturally and adversarially fine-tuned CLIP: (a) sign agreement and (b) cosine similarity for local gradients and neighborhood gradients (aggregated within an $\ell_2$-ball) across perturbation radius. Alignment degrades with attack strength, motivating selective (not uniform) parameter merging.

Through comprehensive experiments, we demonstrate that our PISTOLE consistently achieves state-of-the-art trade-offs between natural performance and robustness compared to existing methods across diverse datasets, architectures, and scenarios. Furthermore, we show that our obtained accuracy-robustness trade-off effectively transfers to a spectrum of downstream vision-language tasks, including captioning, visual question answering, hallucination mitigation, and reasoning, simply through a plug-and-play replacement of the vision encoder with the robustly merged encoder.

Our core contributions are summarized as follows:

1. We systematically explore the feasibility of parameter-level merging conflicting objectives (natural performance and robustness) via empirical gradient analyses.

2. To address this trade-off without costly fine-tuning, we introduce PISTOLE, a novel prediction stability-aware model merging framework that leverages gradient-informed stability masks and multi-step adversarial parameter perturbations for precise parameter re-weighting.

3. We provide theoretical analyses proving that PISTOLE identifies parameter-sensitive predictions in high-curvature regions, guiding selective merging for improved accuracy-robustness trade-offs.

4. We conduct extensive experiments to demonstrate the efficacy and generalizability of PISTOLE across tasks and scenarios, scaling without incurring additional fine-tuning costs.

## 2 RELATED WORKS

**Trade-offs in foundation VLMs.** Foundation VLMs (*e.g.*, CLIP (Radford et al., 2021), LLaVA (Liu et al., 2024), OpenFlamingo (Awadalla et al., 2023)) achieve strong zero-shot transfer via large-scale image–text pre-training, yet core tensions constrain practical deployment: size *vs.* efficiency (Vasu et al., 2025), specialization *vs.* generalization (Zang et al., 2024), and fairness (Luo et al., 2024). Among these, the trade-off between adversarial robustness and natural performance remains particularly challenging, as gains in adversarial robustness often degrade clean performance, consistent with theory on competing objectives (Zhang et al., 2019). In this work, we target this trade-off in foundation VLMs, seeking to enhance robustness while preserving much natural performance.

**Parameter-space model merging.** Beyond prediction ensembles (Yang et al., 2023), parameter merging combines knowledge from VLMs without retraining (Wortsman et al., 2022b;a). A key approach leverages task vectors (*i.e.*, parameter differences between fine-tuned and pre-trained VLMs) (Ilharco et al., 2023; Ortiz-Jimenez et al., 2023), providing a flexible mechanism for merging knowledge. Recent works like Ties-Merging (Yadav et al., 2023) proposed resolving interference between merged models by identifying parameter conflicts, while AdaMerging (Yang et al., 2024) introduced adaptive merging strategies for multi-task learning. These methods overlook parameter-space perturbations and local loss geometry. We thus complement this line by using gradient-informed masks and adversarial parameter trajectories to account for sensitivity and curvature during merging.

**Adversarial robustness of foundation VLMs.** Adversarial robustness in VLMs remains a critical problem, with recent works (Mao et al., 2023; Schlarmann et al., 2024) mainly pursuing adversarial fine-tuning by integrating adversarial examples into training to bolster robustness. However, these methods often erode clean accuracy and require heavy hyperparameter tuning and costly retraining, limiting scalability. We instead merge off-the-shelf naturally and robustly fine-tuned VLMs via task vectors, using gradient-informed stability masks and adversarial parameter trajectories to trace gradient paths during merging, reconciling accuracy and robustness without additional fine-tuning.

## 3 PREDICTION STABILITY-AWARE MODEL MERGING

Below, we propose **PISTOLE**, the first task vector-based model merging method to address the accuracy-robustness trade-off without costly adversarial fine-tuning, generalizing across tasks.

### 3.1 *Background*: ADVERSARIAL FINE-TUNING AND TASK VECTORS

**CLIP.** As a milestone of multimodal learning, CLIP (Radford et al., 2021) employs two modality-specific encoders: an image encoder $f_{\boldsymbol{\theta}_{\mathrm{I}}}: \mathcal{X} \to \mathbb{R}^d$ and a text encoder $f_{\boldsymbol{\theta}_{\mathrm{T}}}: \mathcal{T} \to \mathbb{R}^d$, whose outputs reside in a shared $d$–dimensional embedding space. For an input image $\mathbf{x}$ and a set of class prompts $\{\mathbf{t}_1, \ldots, \mathbf{t}_C\}$ constructed from templates (*e.g.*, ``This is a photo of [CLASS_$c$]''), the prediction is obtained via the cosine similarity between the visual feature and each textual feature:

$$p_c(\mathbf{x}; \boldsymbol{\theta}_{\mathrm{I}}, \boldsymbol{\theta}_{\mathrm{T}}) = \frac{\exp\big(\cos(f_{\boldsymbol{\theta}_{\mathrm{I}}}(\mathbf{x}), f_{\boldsymbol{\theta}_{\mathrm{T}}}(\mathbf{t}_c))\big)}{\sum_{c'=1}^{C} \exp\big(\cos(f_{\boldsymbol{\theta}_{\mathrm{I}}}(\mathbf{x}), f_{\boldsymbol{\theta}_{\mathrm{T}}}(\mathbf{t}_{c'}))\big)}, \tag{1}$$

where $\exp(\cdot)$ denotes the exponential function, and $\cos(\cdot, \cdot)$ computes the cosine similarity between two embeddings that have been $\ell_2$ normalized. The prediction vector w.r.t. the CLIP parameter set $\boldsymbol{\theta} = [\boldsymbol{\theta}_{\mathrm{I}}, \boldsymbol{\theta}_{\mathrm{T}}]$ across $C$ categories is written as $\mathbf{p}_{\boldsymbol{\theta}}(\mathbf{x}) = [p_1(\mathbf{x}; \boldsymbol{\theta}_{\mathrm{I}}, \boldsymbol{\theta}_{\mathrm{T}}), \ldots, p_C(\mathbf{x}; \boldsymbol{\theta}_{\mathrm{I}}, \boldsymbol{\theta}_{\mathrm{T}})]^\top$.

**Standard adversarial fine-tuning (TeCoA).** In consistent with the standard adversarial training paradigm (Madry et al., 2018), TeCoA (Mao et al., 2023) enhances CLIP robustness against $\ell_\infty$-norm adversarial attacks by solving the following minimax optimization problem:

$$\min_{\boldsymbol{\theta}_{\mathrm{I}}} \mathbb{E}_{(\mathbf{x},c)\sim\mathcal{D}}\left[\max_{\|\boldsymbol{\delta}\|_\infty \leq \epsilon} \mathcal{L}_{\mathrm{CE}}\big(\mathbf{p}_{\boldsymbol{\theta}}(\mathbf{x} + \boldsymbol{\delta}), \mathbf{e}_c\big)\right], \tag{2}$$

where $\mathbf{e}_c = [\mathbb{1}(c=1), \ldots, \mathbb{1}(c=C)]^\top \in \{0,1\}^C$ is the one-hot label for class $c$, and $\mathcal{L}_{\mathrm{CE}}$ denotes the cross-entropy loss. The inner maximization is approximated by the $m$-step Projected Gradient Descent (PGD) (Calamai & Moré, 1987) on the negative loss function:

$$\hat{\mathbf{x}}^{(i+1)} = \Pi_{\mathbb{B}(\mathbf{x},\epsilon)}\left[\hat{\mathbf{x}}^{(i)} + \alpha \cdot \mathrm{sign}\left(\nabla_{\hat{\mathbf{x}}^{(i)}} \mathcal{L}_{\mathrm{CE}}\big(\mathbf{p}_{\boldsymbol{\theta}}(\hat{\mathbf{x}}^{(i)}), \mathbf{e}_c\big)\right)\right], \tag{3}$$

initialized with $\hat{\mathbf{x}}^{(0)} = \mathbf{x} + 0.001 \cdot \mathcal{N}(\mathbf{0}, \mathbf{I})$. Here, $\alpha$ represents the step size, $\mathrm{sign}(\cdot)$ is the element-wise sign function, and $\Pi_{\mathbb{B}(\mathbf{x},\epsilon)}(\cdot)$ denotes the projection onto the $\ell_\infty$ ball of radius $\epsilon$. Further details of other adversarial fine-tuning approaches are in Appendix B.

**Task vectors for multi-task adaptation.** Given a downstream task $\mathcal{T}_i$ with data $\mathcal{D}_i$, fine-tuning a pre-trained VLM $\boldsymbol{\theta}_0$ yields task-specific parameters $\boldsymbol{\theta}_i$ and the *task vector* (Ilharco et al., 2023) (parameter displacement) $\boldsymbol{\tau}_i = \boldsymbol{\theta}_i - \boldsymbol{\theta}_0$. Task vectors compose for vision encoders: for $\{\boldsymbol{\tau}_i\}_{i=1}^M$, the aggregate $\boldsymbol{\tau}_{\text{add}} = \sum_{i=1}^M \boldsymbol{\tau}_i$ defines the merged model $\boldsymbol{\theta}_{\text{add}} = \boldsymbol{\theta}_0 + \lambda \cdot \boldsymbol{\tau}_{\text{add}}$, with the scalar $\lambda$ tuned on a validation set. This simple addition typically attains competitive performance across the $M$ tasks.

## 3.2 CAN VANILLA TASK-VECTOR ADDITION AID TRADE-OFF?

**Weight-space interpolation reveals an InD–shift sweet point.** Following WiSE-FT (Wortsman et al., 2022b), let $\boldsymbol{\theta}_0$ denote a pre-trained zero-shot VLM and $\boldsymbol{\theta}_{\text{FT}}$ its VLM fine-tuned on a *reference* (in-distribution, InD) dataset, *e.g.*, ImageNet. Evaluating on both the reference distribution and *shifted* distributions (*e.g.*, natural variants and subpopulations), the linear interpolation $\boldsymbol{\theta}_{\text{WISE}}(\lambda) = (1 - \lambda)\,\boldsymbol{\theta}_0 + \lambda\,\boldsymbol{\theta}_{\text{FT}}, \lambda \in [0, 1]$, typically traces a Pareto-like curve with an interior $\lambda^\star$ that preserves high InD accuracy while improving accuracy under distribution shift (see Figure 2). This simple weight-space averaging with a sweet point (improved trade-off) motivates viewing parameter operations (including task-vector arithmetic) as a light-weight alternative to exhaustive hyperparameter

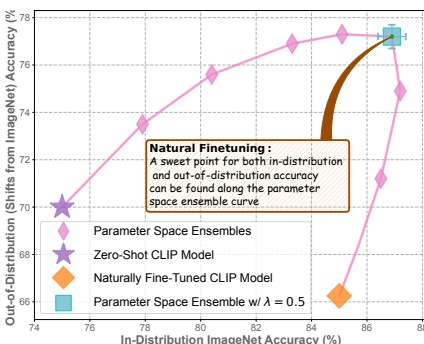

Figure 2: Parameter-space ensembling of **pre-trained and fine-tuned CLIP.**

sweeps when balancing specialization to the reference data against generalization to shifted data.

**Vanilla addition fails to balance natural performance and adversarial robustness.** Let $\boldsymbol{\theta}_{\text{nat}}$ and $\boldsymbol{\theta}_{\text{rob}}$ be naturally and robustly (adversarially) fine-tuned models with task vectors $\boldsymbol{\tau}_{\text{nat}} = \boldsymbol{\theta}_{\text{nat}} - \boldsymbol{\theta}_0$ and $\boldsymbol{\tau}_{\text{rob}} = \boldsymbol{\theta}_{\text{rob}} - \boldsymbol{\theta}_0$. Under the naive interpolation $\boldsymbol{\theta}_{\text{van}}(\lambda) = \boldsymbol{\theta}_0 + (1 - \lambda)\,\boldsymbol{\tau}_{\text{nat}} + \lambda\,\boldsymbol{\tau}_{\text{rob}}, \lambda \in [0, 1]$, clean and adversarial accuracies vary almost *linearly* and antagonistically, producing a near straight line between endpoints with no pronounced interior optimum (Figure 3). The issue is equal weighting: it ignores which coordinates align or conflict across objectives, so robustness gains come by eroding clean accuracy at a roughly constant rate. Gradient analyses (Figure 1) indicate that compatible and conflicting directions co-exist, motivating a prediction stability–aware, selectively re-weighted

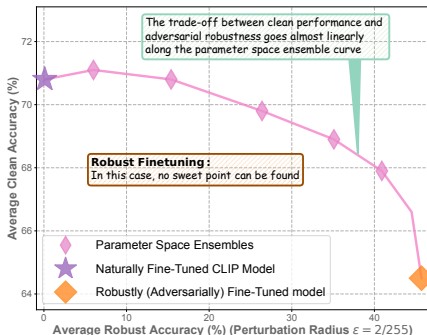

Figure 3: Parameter ensembling of **naturally and robustly fine-tuned CLIP.**

merge that preserves consensus coordinates while attenuating counterpart-sensitive ones.

## 3.3 GRADIENT-INFORMED STABILITY RE-WEIGHTING

Vanilla task vector merging typically treats all coordinates equally, reproducing the near-linear clean–robust trade-off of $\boldsymbol{\theta}_{\text{van}}(\lambda)$. In contrast, Figure 1 shows only modest alignment that degrades with attack strength, indicating a mix of compatible and conflicting coordinates across the natural and adversarial objectives. We therefore build *complementary masks* that (i) preserve coordinates the counterpart objective deems stable and (ii) attenuate coordinates where the counterpart exhibits large gradient magnitude (*i.e.*, it would strongly update those weights in an opposing direction).

**Aggregated gradients and layer-wise scaling.** Raw per-batch gradients are noisy and differ in scale across layers; without normalization, a few high-variance tensors dominate the mask. We first accumulate expected gradients for the two objectives and then normalize them per layer:

$$\mathbf{g}_{\text{nat}} = \mathbb{E}_{(\mathbf{x}, c) \sim \mathcal{D}}\big[\nabla_{\boldsymbol{\theta}_{\text{nat}}} \mathcal{L}_{\text{CE}}(\mathbf{p}_{\boldsymbol{\theta}_{\text{nat}}}(\mathbf{x}), \mathbf{e}(c))\big], \qquad \mathbf{g}_{\text{rob}} = \mathbb{E}_{(\hat{\mathbf{x}}, c) \sim \hat{\mathcal{D}}}\big[\nabla_{\boldsymbol{\theta}_{\text{rob}}} \mathcal{L}_{\text{CE}}(\mathbf{p}_{\boldsymbol{\theta}_{\text{rob}}}(\hat{\mathbf{x}}), \mathbf{e}(c))\big], \quad (4)$$

where $\hat{\mathbf{x}}$ is an adversarial sample for $\boldsymbol{\theta}_{\text{rob}}$ obtained by Eq. (3). For each objective index $s \in \{\text{nat}, \text{rob}\}$ and each layer $l$, let $\mathbf{g}_s^{(l)}$ denote the gradient tensor of layer $l$. We define the per-layer normalization:

$$\text{Norm}\big(\mathbf{g}_s^{(l)}\big) = \frac{|\mathbf{g}_s^{(l)}|}{\max(|\mathbf{g}_s^{(l)}|) + \varepsilon} \in [0, 1]^{\text{shape}(\mathbf{g}_s^{(l)})}, \qquad \tilde{\mathbf{g}}_s^{(l)} = \text{Norm}\big(\mathbf{g}_s^{(l)}\big)^\gamma, \quad (5)$$

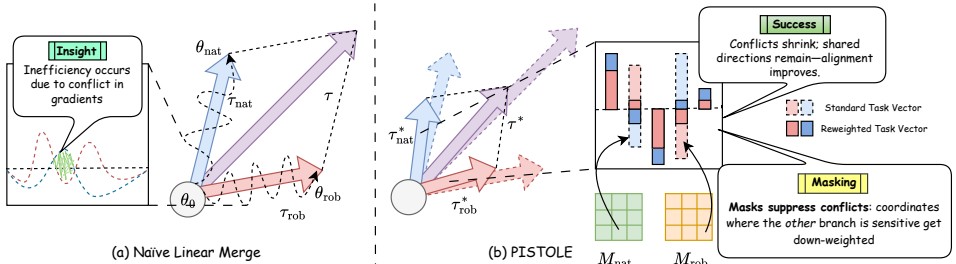

Figure 4: (a) Naive linear merge uniformly adds natural and robust task vectors, ignoring conflicts and yielding a near-linear trade-off. (b) **PISTOLE** merges with complementary, gradient-informed masks, suppressing conflicts and preserving shared directions for a better clean-robust balance.

where small perturbation $\varepsilon > 0$ and temperature $\gamma \in [0, 1]$ are to control dynamic range. Stacking across layers yields $\tilde{\mathbf{g}}_{\text{nat}} = \text{concat}_l \, \tilde{\mathbf{g}}_{\text{nat}}^{(l)}$ and $\tilde{\mathbf{g}}_{\text{rob}} = \text{concat}_l \, \tilde{\mathbf{g}}_{\text{rob}}^{(l)}$, both in $[0, 1]^d$.

**Complementary stability masks.** Gradients generally indicate which coordinates each objective would change, which means that large magnitudes flag parameters that are *sensitive* for that objective. To avoid reintroducing antagonism during merging, we suppress coordinates that the *other* fine-tuning objective wishes to change most, while preserving those it considers stable. Concretely, for $s \in \{\text{nat}, \text{rob}\}$ and each layer $l$, let $\tilde{\mathbf{g}}_s^{(l)} \in [0, 1]^{\text{shape}}$ be the normalized gradient magnitudes from Eq. (5), and write $\tilde{\mathbf{g}}_s = \text{concat}_l \, \tilde{\mathbf{g}}_s^{(l)} \in [0, 1]^d$. We convert these into complementary masks:

$$\mathbf{M}_{\text{nat}} = \left(\mathbf{1} - \tilde{\mathbf{g}}_{\text{rob}}\right)^{\kappa}, \qquad \mathbf{M}_{\text{rob}} = \left(\mathbf{1} - \tilde{\mathbf{g}}_{\text{nat}}\right)^{\kappa}, \tag{6}$$

where $\kappa \geq 1$ sharpens selectivity. To guarantee a user-controlled stability budget, we apply a per-layer quantile cap, $\mathbf{M}_{\text{nat},l} \leftarrow \min\left(\mathbf{M}_{\text{nat},l}, t_{q,l}^{\text{rob}}\right)$ and $\mathbf{M}_{\text{rob},l} \leftarrow \min\left(\mathbf{M}_{\text{rob},l}, t_{q,l}^{\text{nat}}\right)$, where $t_{q,l}^{\text{rob}}$ (*resp.*, $t_{q,l}^{\text{nat}}$) is the $q$-quantile of $\mathbf{M}_{\text{nat},l}$ (*resp.*, $\mathbf{M}_{\text{rob},l}$). This attenuates the layerwise top-$q$ most counterpart-sensitive coordinates, thereby upper-bounding first-order cross-objective interference.

**Theorem 1.** *Let* $\mathbf{g}_{nat}^0 := \nabla_{\boldsymbol{\theta}} \mathcal{L}_{nat}(\boldsymbol{\theta})\big|_{\boldsymbol{\theta}=\boldsymbol{\theta}_0}$ *and* $\mathbf{g}_{rob}^0 := \nabla_{\boldsymbol{\theta}} \mathcal{L}_{rob}(\boldsymbol{\theta})\big|_{\boldsymbol{\theta}=\boldsymbol{\theta}_0}$. *Let* $\mathbf{M}_{nat}^{pre} := \left(\mathbf{1} - \tilde{\mathbf{g}}_{rob}\right)^{\kappa}$ *and* $\mathbf{M}_{rob}^{pre} := \left(\mathbf{1} - \tilde{\mathbf{g}}_{nat}\right)^{\kappa}$ *be the uncapped complementary masks from Eq. (6), with* $\kappa \geq 1$ *and* $\tilde{\mathbf{g}}_s \in [0, 1]^d$ *defined by Eq. (5). For each layer* $l$, *let* $t_{q,l}^{rob}$ *be the* $q$-quantile of $(\mathbf{M}_{nat}^{pre})_l$ *and* $t_{q,l}^{nat}$ *the* $q$-quantile of $(\mathbf{M}_{rob}^{pre})_l$ *(empirical quantiles on layer entries).[1] Define the capped masks layerwise by* $(\mathbf{M}_{nat})_l := \min\left((\mathbf{M}_{nat}^{pre})_l, t_{q,l}^{rob}\mathbf{1}\right)$ *and* $(\mathbf{M}_{rob})_l := \min\left((\mathbf{M}_{rob}^{pre})_l, t_{q,l}^{nat}\mathbf{1}\right)$. *Set* $\rho_{nat} := \max_l t_{q,l}^{nat}$ *and* $\rho_{rob} := \max_l t_{q,l}^{rob}$. *Then for any* $\boldsymbol{\delta} \in \mathbb{R}^d$,

$$\left|\langle \mathbf{g}_{nat}^0, \mathbf{M}_{rob} \odot \boldsymbol{\delta} \rangle\right| \leq \rho_{nat} \, \|\mathbf{g}_{nat}^0\|_2 \, \|\boldsymbol{\delta}\|_2, \qquad \left|\langle \mathbf{g}_{rob}^0, \mathbf{M}_{nat} \odot \boldsymbol{\delta} \rangle\right| \leq \rho_{rob} \, \|\mathbf{g}_{rob}^0\|_2 \, \|\boldsymbol{\delta}\|_2. \tag{7}$$

*Moreover, if* $\kappa$ *is increased (i.e., sharpening* $(1 - \tilde{\mathbf{g}})^{\kappa}$) *or any of the layerwise caps* $t_{q,l}^{nat}, t_{q,l}^{rob}$ *are decreased, the right-hand sides in Eq. (7) are monotone nonincreasing.*

*Proof.* The full proof is provided in Appendix D.1. $\square$

**Corollary 1** (Worst-case first-order contraction *vs.* uniform addition). *For task vectors* $\boldsymbol{\tau}_{nat} = \boldsymbol{\theta}_{nat} - \boldsymbol{\theta}_0$ *and* $\boldsymbol{\tau}_{rob} = \boldsymbol{\theta}_{rob} - \boldsymbol{\theta}_0$, $\left|\langle \mathbf{g}_{nat}^0, \mathbf{M}_{rob} \odot \boldsymbol{\tau}_{rob} \rangle\right| \leq \rho_{nat} \|\mathbf{g}_{nat}^0\|_2 \|\boldsymbol{\tau}_{rob}\|_2 \leq \|\mathbf{g}_{nat}^0\|_2 \|\boldsymbol{\tau}_{rob}\|_2$, *and the symmetric bound holds swapping* $(nat, rob)$. *Hence, complementary masking contracts worst-case cross-objective first-order interference by factors* $\rho_{nat}, \rho_{rob} \leq 1$ *relative to unmasked mixing.*

> Theorem 1 formalizes a first-order non-interference guarantee: when any displacement is filtered by the counterpart's mask, the *first-order* increase of the other objective's loss is bounded by a tunable multiplicative factor $\rho \leq 1$. The factors $\rho_{\text{nat}}$ and $\rho_{\text{rob}}$ depend only on layerwise caps (through their maxima) and respond monotonically: larger $\kappa$ (sharper masks) or tighter caps reduce $\rho$ and thus strengthen attenuation. Practically, this means masked combinations curve the otherwise near-linear clean–robust trade-off of uniform addition by suppressing coordinates that the counterpart objective marks as sensitive, while preserving jointly stable coordinates.

---

[1] Any standard definition of the empirical $q$-quantile with $q \in (0, 1]$ suffices. We here consider that quantiles are monotone under component-wise decreases.

## 3.4 TRACING ADVERSARIAL PATHS IN PARAMETER SPACE

The complementary masks in Section 3.3 are built from *single-point* gradient magnitudes at $(\boldsymbol{\theta}_{\text{nat}}, \boldsymbol{\theta}_{\text{rob}})$, which capture first-order instability but can miss nearby high-curvature pockets where sensitivity spikes. To refine these stability estimates without retraining, we augment them with *adversarial parameter perturbation*: we trace short parameter updating trajectories in a small neighborhood of each fine-tuned solution and aggregate gradients *along* these paths. Intuitively, coordinates that remain stable under worst-direction parameter nudges are safe to keep, whereas coordinates of large gradients along these trajectories are fragile and should be attenuated during merging.

**Adversarial parameter trajectories.** For each objective $s \in \{\text{nat}, \text{rob}\}$, define the Frobenius ball $\mathcal{V}_{\boldsymbol{\theta}_s} = \{\boldsymbol{\Delta} \colon \|\boldsymbol{\Delta}\|_F \leq \eta \|\boldsymbol{\theta}_s\|_F\}$ with radius factor $\eta > 0$. Starting from $\boldsymbol{\theta}_s^{(0)} = \boldsymbol{\theta}_s$, we perform $K$ projected gradient ascent steps in *parameter space* that follow locally worst-case directions:

$$\boldsymbol{\theta}_s^{(i+1)} \ \leftarrow \ \Pi_{\boldsymbol{\theta}_s + \mathcal{V}_{\boldsymbol{\theta}_s}}\big(\boldsymbol{\theta}_s^{(i)} + \beta\,\mathbf{u}_s^{(i)}\big), \qquad \mathbf{u}_s^{(i)} \ := \ \frac{\nabla_{\boldsymbol{\theta}}\mathcal{L}_s\big(\mathbf{p}_{\boldsymbol{\theta}}(\mathbf{x}_s), \mathbf{e}(c)\big)\big|_{\boldsymbol{\theta}=\boldsymbol{\theta}_s^{(i)}}}{\left\|\nabla_{\boldsymbol{\theta}}\mathcal{L}_s\big(\mathbf{p}_{\boldsymbol{\theta}}(\mathbf{x}_s), \mathbf{e}(c)\big)\big|_{\boldsymbol{\theta}=\boldsymbol{\theta}_s^{(i)}}\right\|_F + \epsilon}, \tag{8}$$

where $\beta > 0$ is the step size, and $\Pi_{\boldsymbol{\theta}_s + \mathcal{V}_{\boldsymbol{\theta}_s}}(\cdot)$ projects onto $\mathcal{V}_{\boldsymbol{\theta}_s}$. We take $\mathbf{x}_{\text{nat}} = \mathbf{x}$ (clean inputs) and $\mathbf{x}_{\text{rob}} = \hat{\mathbf{x}}$ (adversarial inputs), so that $\mathcal{L}_{\text{nat}}$ and $\mathcal{L}_{\text{rob}}$ are evaluated under their respective input regimes.

**Path-integrated gradients.** To calibrate the path-integrated gradients with local sensitivity, we introduce the *Prediction Criticality Index* (PCI): a curvature-aware scalar that measures how fragile a prediction is to small *parameter*-space perturbations. We use PCI to weight steps along the adversarial parameter trajectories, so high-curvature (fragile) regions contribute more to the accumulated gradient, while flat, confidence-saturated regions are deemphasized. Formally:

**Definition 1** (Prediction Criticality Index (PCI)). *Let $\boldsymbol{\theta} \in \mathbb{R}^d$ be the model parameters and $\mathbf{p}(\mathbf{x}; \boldsymbol{\theta}) \in [0, 1]^C$ be the prediction for an input $\mathbf{x}$. For a fixed radius factor $\eta > 0$, define the parameter-level Frobenius perturbation ball $\mathcal{V}_{\boldsymbol{\theta}} := \{\boldsymbol{\Delta} \in \mathbb{R}^d : \|\boldsymbol{\Delta}\|_F \leq \eta \|\boldsymbol{\theta}\|_F\}$. Given $\boldsymbol{\Delta}$ sampled isotropically and with zero mean from $\mathcal{V}_{\boldsymbol{\theta}}$ (e.g., uniform in the hyperball). Its covariance is denoted as $\sigma^2 \mathbf{I}_d$, where the per-coordinate second moment is $\sigma^2 := \frac{1}{d}\mathbb{E}_{\boldsymbol{\Delta} \in \mathcal{V}_{\boldsymbol{\theta}}}[\|\boldsymbol{\Delta}\|_2^2] = \frac{\eta^2 \|\boldsymbol{\theta}\|_F^2}{d}$. The equality on the right follows directly from the radius normalization for any isotropic, zero-mean law supported on the ball $\mathcal{V}_{\boldsymbol{\theta}}$. For the ground-truth class $c \in \{1, \ldots, C\}$, we define PCI as follows:*

$$PCI(\mathbf{x}, c, \boldsymbol{\theta}) := \left|\mathbb{E}_{\boldsymbol{\Delta} \in \mathcal{V}_{\boldsymbol{\theta}}}\Big[\frac{\mathbf{p}_c(\mathbf{x}; \boldsymbol{\theta} + \boldsymbol{\Delta}) - \mathbf{p}_c(\mathbf{x}; \boldsymbol{\theta})}{\mathbf{p}_c(\mathbf{x}; \boldsymbol{\theta})}\Big]\right|. \tag{9}$$

A *large* PCI w.r.t. clean/adversarial examples indicates that prediction confidence is highly sensitive to small parameter changes (fragile knowledge), while a small value reflects robustness. We therefore accumulate *path-integrated* gradients:

$$\mathbf{G}_s \ := \ \mathbb{E}_{(\mathbf{x},c) \sim \mathcal{D}_s}\left[\sum_{i=0}^{K} \text{PCI}(\mathbf{x}_s, c, \boldsymbol{\theta})\, \nabla_{\boldsymbol{\theta}}\mathcal{L}_s\big(\mathbf{p}_{\boldsymbol{\theta}}(\mathbf{x}_s), \mathbf{e}(c)\big)\Big|_{\boldsymbol{\theta}=\boldsymbol{\theta}_s^{(i)}}\right], \qquad s \in \{\text{nat}, \text{rob}\}. \tag{10}$$

We then normalize per layer as in Eq. (5) to obtain scores $\tilde{\mathbf{g}}_s^{\text{path}} \in [0, 1]^d$, and form *path-refined complementary masks* via $\mathbf{M}_{\text{nat}}^{\text{path}} = \big(\mathbf{1} - \tilde{\mathbf{g}}_{\text{rob}}^{\text{path}}\big)^{\kappa}$, and $\mathbf{M}_{\text{rob}}^{\text{path}} = \big(\mathbf{1} - \tilde{\mathbf{g}}_{\text{nat}}^{\text{path}}\big)^{\kappa}$, followed by the same per-layer caps as in Section 3.3. The final merged displacement keeps the stable parts:

$$\boldsymbol{\tau}^*(\lambda) \ = \ \lambda\big(\mathbf{M}_{\text{nat}}^{\text{path}} \odot \boldsymbol{\tau}_{\text{nat}}\big) + (1 - \lambda)\big(\mathbf{M}_{\text{rob}}^{\text{path}} \odot \boldsymbol{\tau}_{\text{rob}}\big), \qquad \boldsymbol{\theta}_{\text{PISTOLE}}(\lambda) = \boldsymbol{\theta}_0 + \boldsymbol{\tau}^*(\lambda). \tag{11}$$

Empirically, varying $\lambda$ with $\boldsymbol{\theta}_{\text{PISTOLE}}(\lambda)$ bends the clean–robust frontier beyond the near-linear trade-off of uniform addition, yielding interior points that outperform naive mixing. Figure 4 is an overview of our PISTOLE method compared with naive linear merging. See Appendix E for pseudocode. We next formalize the curvature link that motivates our PCI-based weighting.

**Theorem 2.** *Let PCI be defined as in Definition 1, where $\mathbf{p}_c(\mathbf{x}; \boldsymbol{\theta}) > 0$ is twice continuously differentiable in a neighborhood of $\boldsymbol{\theta}$. Given $\mathbf{H}_c(\boldsymbol{\theta}) := \nabla_{\boldsymbol{\theta}}^2 \mathbf{p}_c(\mathbf{x}; \boldsymbol{\theta})$, for sufficiently small $\eta$, we have the following approximation:*

$$\text{PCI}(\mathbf{x}, c, \boldsymbol{\theta}) = \frac{\sigma^2}{2}\frac{\text{Tr}(\mathbf{H}_c(\boldsymbol{\theta}))}{\mathbf{p}_c(\mathbf{x}; \boldsymbol{\theta})} + \mathcal{O}(\sigma^3). \tag{12}$$

Table 1: Zero-shot accuracy of diverse adversarial learning methods evaluated on 14 datasets. Metrics: **Clean**, **Robust** (AutoAttack, $\ell_\infty$-norm $\epsilon = 2/255$) **Accuracy**, and Clean+Robust **Sum**.

| Eval. | Method | ImageNet | STL10 | CIFAR-10 | CIFAR-100 | Stanf.Cars | Caltech101 | OxfordPet | Flower102 | DTD | EuroSAT | FGVC | PCAM | ImageNet-R | ImageNet-S | Average |
|---|---|---|---|---|---|---|---|---|---|---|---|---|---|---|---|---|
| **Clean** | Standard CLIP | 74.90 | 99.31 | 95.20 | 71.08 | 77.91 | 83.29 | 93.21 | 79.17 | 55.21 | 62.65 | 31.77 | 52.01 | 87.86 | 59.61 | 73.08 |
| | TeCoA | 80.00 | 95.40 | 86.88 | 61.64 | 44.45 | 80.33 | 80.78 | 51.83 | 45.43 | 23.48 | 15.00 | 58.39 | 79.40 | 58.77 | 61.56 |
| | PMG | 77.84 | 96.92 | 90.25 | 64.97 | 58.23 | 83.34 | 86.45 | 58.46 | 46.49 | 28.04 | 20.64 | 49.99 | 83.18 | 57.62 | 64.46 |
| | FARE | 72.96 | 98.28 | 90.24 | 67.78 | 66.80 | 85.65 | 89.75 | 65.13 | 50.43 | 16.54 | 22.83 | 50.02 | 83.75 | 56.86 | 65.50 |
| | TGA | 80.26 | 96.83 | 88.07 | 60.86 | 49.81 | 81.54 | 81.11 | 51.49 | 45.96 | 30.30 | 14.22 | 49.95 | 80.20 | 58.89 | 62.11 |
| | **PISTOLE** | **80.82** | **98.56** | **90.83** | **68.18** | **67.35** | **86.20** | **91.35** | **70.08** | **51.22** | **30.89** | **26.42** | **62.34** | **85.18** | **59.91** | **69.24** |
| **Robust** | Standard CLIP | 0.00 | 0.02 | 0.00 | 0.00 | 0.00 | 0.00 | 0.00 | 0.00 | 0.00 | 0.00 | 0.03 | 0.01 | 0.01 | 0.09 | 0.01 |
| | TeCoA | 61.74 | 86.34 | 61.99 | 35.82 | 18.62 | 70.57 | 68.22 | 27.27 | 26.17 | **12.37** | 5.43 | 26.93 | 59.57 | 44.56 | 43.26 |
| | PMG | 60.02 | 88.21 | 64.12 | 37.14 | 23.68 | 72.47 | 70.92 | 28.20 | 26.33 | 9.07 | 5.79 | 47.06 | 62.24 | 45.08 | 45.74 |
| | FARE | 43.56 | 88.55 | 61.82 | 34.89 | 23.74 | 70.88 | 67.70 | 32.95 | 25.69 | **49.39** | 3.76 | 56.47 | 56.47 | 36.86 | 42.97 |
| | TGA | 61.46 | 88.56 | 63.21 | 35.44 | 21.60 | 71.16 | 68.52 | 26.15 | 26.70 | 11.37 | 5.76 | 47.88 | 60.32 | 44.46 | 45.19 |
| | **PISTOLE** | **61.89** | **89.09** | **66.71** | **39.83** | **28.36** | **72.83** | **71.30** | **33.42** | **28.84** | 11.02 | 6.61 | 48.32 | **63.03** | **45.85** | **47.65** |
| **Sum** | Standard CLIP | 74.90 | 99.33 | 95.20 | 71.08 | 77.91 | 83.29 | 93.21 | 79.17 | 55.21 | 62.65 | 31.80 | 52.02 | 87.87 | 59.70 | 73.09 |
| | TeCoA | 141.74 | 181.74 | 148.87 | 97.46 | 63.07 | 150.90 | 149.00 | 79.10 | 71.60 | 35.85 | 20.43 | 85.32 | 138.97 | 103.33 | 104.82 |
| | PMG | 137.86 | 185.13 | 154.37 | 102.11 | 81.91 | 155.81 | 157.37 | 86.66 | 72.82 | 37.11 | 26.43 | 97.05 | 145.42 | 102.70 | 110.20 |
| | FARE | 116.52 | 186.83 | 152.06 | 102.67 | 90.54 | 156.53 | 157.45 | 98.08 | 76.12 | 20.30 | 28.14 | 99.41 | 140.22 | 93.72 | 108.47 |
| | TGA | 141.72 | 185.39 | 151.28 | 96.30 | 71.41 | 152.70 | 149.63 | 77.64 | 72.66 | 41.67 | 19.98 | 97.83 | 140.52 | 103.35 | 107.30 |
| | **PISTOLE** | **142.71** | **187.65** | **157.54** | **108.01** | **95.71** | **159.03** | **162.65** | **103.50** | **80.06** | **41.91** | **33.03** | **110.66** | **148.21** | **105.76** | **116.89** |

Table 2: Average accuracy (%) of **diverse CLIP backbones** with perturbation radius $\epsilon = 2/255$.

| Backbone | Method | Clean | Robust | Sum |
|---|---|---|---|---|
| ViT-H/14 | TeCoA | 66.95 | 48.63 | 115.58 |
| | PMG | 68.96 | 50.31 | 119.27 |
| | FARE | 70.55 | 50.25 | 120.80 |
| | TGA | 65.91 | 49.14 | 115.05 |
| | **PISTOLE** | **73.61** | **52.45** | **126.06** |
| ViT-B/32 | TeCoA | 48.83 | 25.75 | 74.58 |
| | PMG | 49.71 | 26.98 | 76.69 |
| | FARE | 56.68 | 29.30 | 85.98 |
| | TGA | 57.54 | 31.15 | 88.69 |
| | **PISTOLE** | **60.71** | **33.58** | **94.29** |

Table 3: Avg. accuracy (%) of **diverse $\epsilon$** when fine-tuning and testing w.r.t. CLIP w/ ViT-L.

| Radius | Method | Clean | Robust | Sum |
|---|---|---|---|---|
| $\epsilon = 3/255$ | TeCoA | 58.90 | 38.07 | 96.97 |
| | PMG | 61.72 | 39.40 | 101.12 |
| | FARE | 63.55 | 37.17 | 100.72 |
| | TGA | 59.65 | 38.59 | 98.24 |
| | **PISTOLE** | **65.09** | **40.57** | **105.66** |
| $\epsilon = 4/255$ | TeCoA | 56.25 | 32.53 | 88.78 |
| | PMG | 58.82 | 33.87 | 92.69 |
| | FARE | 60.26 | 32.02 | 92.28 |
| | TGA | 56.76 | 32.86 | 89.62 |
| | **PISTOLE** | **62.37** | **34.94** | **97.31** |

*Proof.* The full proof is provided in Appendix D.2. □

> Theorem 2 reveals that the PCI is large when the prediction confidence lies in a region of large Hessian trace (high curvature), whereas a small PCI characterizes flat, confidence–saturated zones. High-PCI samples (*i.e.*, those most sensitive to parameter perturbations) are up-weighted in the accumulated gradients, directing the merge to address fragile prediction modes that would otherwise dominate post-fusion error. The sensitivity masks suppress features that are unstable in one model while retaining their more robust analogs in the other, fostering a synergistic blend of natural performance and adversarial robustness.

## 4 EXPERIMENTS

In this section, we compare our PISTOLE method with state-of-the-art adversarial fine-tuning approaches across different scenarios and downstream vision-language tasks.

**Datasets.** Task vectors are obtained by natural and adversarial fine-tuning on ImageNet-1k (Deng et al., 2009). We report zero-shot classification on its test set plus 13 additional datasets, and assess transfer on captioning, visual question answering, hallucination, etc. (details in Appendix C.1).

**Implementation details.** Unless stated otherwise, the base VLM is CLIP with a ViT-L/14 encoder, following robust-CLIP practice (Mao et al., 2023; Schlarmann et al., 2024). The natural VLM is trained by ERM on clean data; the robust VLM follows PMG (Wang et al., 2024) with 10-step PGD ($\ell_\infty$, $\epsilon = 2/255$, step $\alpha = 1/255$). We form task vectors ($\tau_{\text{nat}}, \tau_{\text{rob}}$) and merge them via **PISTOLE** with default mixing $\lambda = 0.2$. Zero-shot robustness is measured with AutoAttack (Croce & Hein, 2020). For downstream transfer, we replace the vision encoder in LLaVA-1.5-7B and OpenFlamingo-9B with our merged encoder while keeping other components fixed. Evaluations use adaptive attacks for fairness. Additional configurations appear in Appendix C.2.

### 4.1 MAIN RESULTS ON ZERO-SHOT CLASSIFICATION

**Zero-shot classification.** Table 1 summarizes clean and robust accuracies for CLIP ViT-L/14 on 14 evaluation sets. In addition to reporting each metric separately, we include a scalar trade-off score *Sum* (Clean+Robust) to capture overall performance. PISTOLE delivers the strongest zero-shot results, improving mean clean accuracy by ∼5% and mean robust accuracy by ∼5.8% over state-of-the-art adversarial fine-tuning baselines. On in-distribution ImageNet, our trade-off is marginally better than alternatives, which we attribute to the robust component obtained via PMG (Wang et al.,

Table 5: Zero-shot transfer on **image captioning** (CIDEr score) and **VQA** (accuracy %).

| VLM Type | Method | Image Captioning | | | | | | Visual Question Answering | | | | | | | | |
|---|---|---|---|---|---|---|---|---|---|---|---|---|---|---|---|---|
| | | COCO | | | Flickr30k | | | TextVQA | | | VQAv2 | | | Vizwiz | | |
| | | Clean | Robust | Sum | Clean | Robust | Sum | Clean | Robust | Sum | Clean | Robust | Sum | Clean | Robust | Sum |
| LLaVA 1.5 | Standard CLIP | 112.3 | 2.9 | 115.2 | 74.7 | 1.0 | 75.7 | 34.8 | 0.0 | 34.8 | 74.5 | 0.0 | 74.5 | 39.4 | 2.3 | 41.7 |
| | TeCoA | 96.7 | 45.1 | 141.8 | 55.2 | 24.0 | 79.2 | 23.8 | 12.8 | 36.6 | 66.2 | 35.7 | 101.9 | 42.5 | 29.6 | 72.1 |
| | PMG | 103.1 | 52.8 | 155.9 | 63.2 | 28.4 | 91.6 | 27.6 | 14.0 | 41.6 | 68.4 | 35.1 | 103.5 | 41.0 | 27.6 | 68.6 |
| | FARE | 108.5 | 47.9 | 156.4 | 67.4 | 24.5 | 91.9 | 30.5 | 14.7 | 45.2 | 70.3 | 34.5 | 104.8 | 41.9 | 25.3 | 67.2 |
| | TGA | 101.3 | 50.6 | 151.9 | 61.9 | 27.8 | 89.7 | 27.1 | 14.6 | 41.7 | 67.3 | 35.0 | 102.3 | 42.8 | 28.0 | 70.8 |
| | **PISTOLE** | **110.6** | **54.9** | **165.5** | **72.8** | **30.5** | **103.3** | **33.1** | **16.0** | **49.1** | **73.7** | **36.9** | **110.6** | **44.5** | **32.3** | **76.8** |
| OpenFlamingo | Standard CLIP | 78.8 | 1.5 | 80.3 | 58.7 | 0.6 | 59.3 | 22.3 | 0.0 | 22.3 | 47.7 | 0.0 | 47.7 | 17.7 | 3.3 | 21.0 |
| | TeCoA | 73.0 | 29.6 | 102.6 | 47.4 | 13.7 | 61.1 | 17.3 | 2.4 | 19.7 | 46.1 | 23.8 | 69.9 | 17.6 | 4.0 | 21.6 |
| | PMG | 76.2 | 31.2 | 107.4 | 52.0 | 16.5 | 68.5 | 17.7 | 2.8 | 20.5 | 47.0 | 24.0 | 71.0 | 16.9 | 4.2 | 21.1 |
| | FARE | 77.9 | 32.7 | 110.6 | 53.5 | 15.9 | 69.4 | 18.8 | 2.2 | 21.0 | 46.7 | 21.8 | 68.5 | 17.2 | 3.8 | 21.0 |
| | TGA | 74.2 | 30.5 | 104.7 | 51.8 | 16.0 | 67.8 | 19.0 | 2.7 | 21.7 | 46.2 | 23.6 | 69.8 | 18.0 | 2.5 | 20.5 |
| | **PISTOLE** | **80.7** | **34.0** | **114.7** | **57.2** | **16.9** | **74.1** | **21.2** | **4.1** | **25.3** | **47.8** | **24.9** | **72.7** | **19.4** | **5.7** | **25.1** |

2024) used in the merge. Section 4.3 further examines how substituting different robustly fine-tuned VLM components can alter the relative trends.

**Robustness generality across backbones.** Beyond ViT-L/14, we evaluate our PISTOLE method with ViT-H/14 and ViT-B/32 based on the clean/robust task vectors of the identical CLIP architecture. As summarized in Table 2, the method consistently surpasses previous adversarial VLM learning approaches in both average clean and robust accuracy across the same 14 datasets.

**Robustness across varying perturbation radius.** We further stress-test robustness by increasing the $\ell_\infty$ budget beyond the default $\epsilon = 2/255$, considering $\epsilon \in 3/255, 4/255$ for both fine-tuning and evaluation to ensure parity. Results in Table 3 indicate that our PISTOLE maintains its lead across these stronger threat models in the zero-shot setting.

**Task vector merging with the PEFT extension.** To reduce the high computational cost brought by full parameter fine-tuning, Parameter-Efficient Fine-Tuning (PEFT) strategies (*e.g.*, LoRA (Hu et al., 2022) based on learnable low-rank matrices for efficient adaptation) were typically paired with adversarial fine-tuning. We thus extend prior adversarial VLM learning methods and our task vector-based merging approach with LoRA. We report both clean and robust accuracy associated with

Table 4: Average accuracy (%) w.r.t. different $\epsilon$ for fine-tuning and testing (ViT-L) **w/ LoRA**.

| Radius | Method | Clean | Robust | Sum |
|---|---|---|---|---|
| $\epsilon = 3/255$ | TeCoA | 55.22 | 26.54 | 81.76 |
| | PMG | 56.82 | 27.02 | 83.84 |
| | FARE | 57.84 | 23.85 | 81.69 |
| | TGA | 55.88 | 26.66 | 82.54 |
| | **PISTOLE** | **59.47** | **29.35** | **88.82** |
| $\epsilon = 4/255$ | TeCoA | 49.84 | 19.87 | 69.71 |
| | PMG | 52.08 | 20.07 | 72.15 |
| | FARE | 53.20 | 19.09 | 72.29 |
| | TGA | 51.13 | 19.83 | 70.96 |
| | **PISTOLE** | **55.06** | **21.78** | **76.84** |

their sum for our LoRA-based PISTOLE method against other LoRA-enabled baselines in Table 4. Even under this efficiency regime, our PISTOLE typically achieves the best trade-off.

## 4.2 ZERO-SHOT DOWNSTREAM TASK TRANSFER

**Transfer to image captioning.** We here evaluate zero-shot task transfer to image captioning by swapping the vision encoder in LLaVA and OpenFlamingo with our PISTOLE-merged encoder. Table 5 (Left) reports the CIDEr score (Vedantam et al., 2015) on COCO and Flickr30k. PISTOLE attains the strongest scores on both clean inputs and under adversaries for a better trade-off, outperforming adversarial learning baselines. Qualitative visualizations are in Figure 6 (Appendix F).

**Transfer to Visual Question Answering (VQA).** Table 5 (Right) also summarizes VQA accuracy across three standard benchmarks. PISTOLE consistently increases the sum relative to prior baselines, primarily by delivering sizable zero-shot robustness gains while keeping natural accuracy essentially intact (and even higher on VizWiz). These results indicate a better accuracy–robustness compromise in the zero-shot regime. Qualitative examples are in Figure 7 (Appendix F).

**Transfer to object hallucination.** To probe hallucination (*i.e.*, erroneously recognizing objects that do not exist in inputs) (Sahoo et al., 2024), we adopt the POPE benchmark (Li et al., 2023) with its standard question-sampling protocols (Appendix C.3). Results in Table 6 indicate that PISTOLE consistently lowers halluci-

Table 6: **POPE hallucination** benchmark (F1-score) with ViT-L using three sampling protocols.

| Method | POPE Sampling | | | Avg. Score |
|---|---|---|---|---|
| | Random | Popular | Adversarial | |
| TeCoA | 79.8 | 79.1 | 75.2 | 78.0 |
| PMG | 81.7 | 80.9 | 76.3 | 79.6 |
| FARE | 82.2 | 81.5 | 78.6 | 80.8 |
| TGA | 80.4 | 79.8 | 76.0 | 78.7 |
| **PISTOLE** | **84.6** | **83.7** | **80.8** | **83.0** |

nation rates versus competing adversarial learning schemes. We attribute this to our stability-aware masking, which dampens over-confident, brittle features and favors parameters that remain reliable under perturbations. Qualitative cases are in Figure 8 in Appendix F.

**Transfer to science question answering w/ Chain-of-Thought (CoT).** We assess CoT reasoning on the ScienceQA benchmark (Lu et al., 2022). Across multiple prompting and VLM settings (Appendix C.3), our PISTOLE achieves the best overall accuracy (Table 7), suggesting that stabilizing the vision backbone from diverse knowledge sources improves the robustness of multi-step reasoning. Illustrative examples are in Figure 9 in Appendix F.

Table 7: CoT eval. (Acc.) using science question answering for adversarial VLM learning (ViT-L).

| Method | Temperature | | | Avg. Acc. |
|---|---|---|---|---|
| | 0.0 | 0.1 | 0.2 | |
| TeCoA | 51.4 | 51.6 | 50.0 | 51.0 |
| PMG | 51.9 | 52.0 | 51.6 | 51.8 |
| FARE | 52.5 | 52.2 | 52.4 | 52.4 |
| TGA | 52.1 | 51.9 | 51.8 | 51.9 |
| **PISTOLE** | **54.1** | **53.9** | **54.2** | **54.1** |

### 4.3 FURTHER ANALYSES (WHY PISTOLE IS EFFECTIVE)

In this section, we conduct a series of controlled ablations of our PISTOLE method and its components to justify its effectiveness and generalizability across different scenarios.

**Effect of PISTOLE components.** We here quantify the contributions of three core modules in our PISTOLE: (i) Gradient-Informed Stability Mask (GISM) in Eq. (6), (ii) Re-weighting of Prediction Criticality Index (PCI) in Definition 1, and (iii) Adversarial Parameter Trajectory (APT) in Eq. (8). Table 8 reports zero-shot results averaged over 14 classification benchmarks. As a reference, *the first row (baseline) applies vanilla task vector merging* in $\theta_{van}(\cdot)$ in Section 3.2. Enforcing stability masks simultaneously improves natural performance and robustness. Augmenting it with PCI further lifts the trade-off by prioritizing fragile predictions. Incorporating adversarial parameter trajectory improves robustness by refining sensitivity estimates beyond single-point gradients.

Table 8: Component ablations for PISTOLE. We report mean clean and robust accuracy (%) averaged on 14 datasets.

| | GISM | PCI | APT | Clean | Robust | Sum |
|---|---|---|---|---|---|---|
| 1 | | | | 66.57 | 44.54 | 111.11 |
| 2 | ✓ | | | 67.78 | 45.69 | 113.47 |
| 3 | ✓ | ✓ | | 68.36 | 46.47 | 114.83 |
| 4 | ✓ | | ✓ | 67.64 | 47.11 | 114.75 |
| 4 | ✓ | ✓ | ✓ | 69.24 | 47.65 | 116.89 |

**Effect of natural knowledge components.** We vary the source of the "natural" task vector while fixing the robust task vector in our PISTOLE method. As shown in Table 9, replacing the pre-trained (zero-shot) VLM with a naturally fine-tuned one yields a stronger clean-robust trade-off. Intuitively, natural empirical risk minimization contributes task-calibrated shifts that better align with the robust objective's consensus direction, which our stability masks preserve while suppressing antagonistic coordinates.

Table 9: Avg. performance (%) of our PISTOLE with diverse **task vectors for natural knowledge**.

| Natural Knowledge Source | Clean | Robust | Sum |
|---|---|---|---|
| Pre-Trained (Zero-Shot) VLM | 67.10 | 47.83 | 114.93 |
| Naturally Fine-tuned VLM | **69.24** | **47.65** | **116.89** |

**Effect of robust knowledge component.** We ablate the source of the robust task vector while fixing the natural one, instantiating PISTOLE with diverse adversarial fine-tuning methods (summarized in Appendix B). From Table 10, we observe a consistent pattern: TeCoA yields the strongest in-distribution (ImageNet) robustness but transfers less favorably under shift, while FARE improves OOD robustness yet lags on ImageNet. Our setup offers the most balanced performance, producing the best trade-off. We attribute this to its prediction-regularized objective, which preserves features aligned with the natural objective. Our stability masks retain these while suppressing antagonistic coordinates.

Table 10: Avg. performance (%) of our PISTOLE with diverse **task vectors for robust knowledge**.

| Robust Knowledge Source | ImageNet | | | Avg. 13 Datasets | | |
|---|---|---|---|---|---|---|
| | Clean | Robust | Sum | Clean | Robust | Sum |
| TeCoA | 79.23 | **62.31** | 141.54 | 64.94 | 43.52 | 108.46 |
| FARE | 75.86 | 62.02 | 137.88 | **69.47** | 44.91 | 114.38 |
| PMG (Our Setup) | **80.82** | 61.89 | **142.71** | 68.35 | 46.55 | **114.90** |

**Curvature analyses.** We quantify loss-parameter curvature along the local update (gradient) direction for adversarial inputs in Figure 5, which plots this directional curvature while changing the merging weight $\lambda$. Vanilla mixing exhibits consistently higher curvature, especially at a larger radius, whereas PISTOLE lowers curvature throughout, indicating a smoother, more stable landscape under parameter nudges. We observe that $\lambda = 0.2$ (our operating point) attains the lowest curvature and simultaneously yields the best trade-off. This behavior aligns with our theorems: PCI weighting prioritizes high-curvature regions (Theorem 2), and the complementary masks provably contract cross-objective first-order interference (Theorem 1), jointly steering the merge toward flatter, better-generalizing solutions.

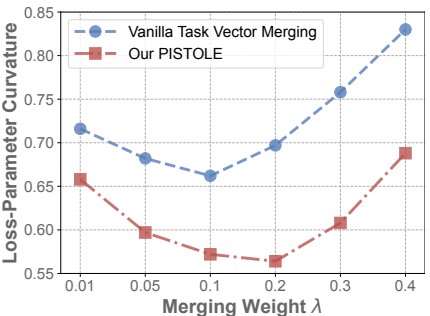

Figure 5: Analyses of loss-parameter curvature on ImageNet adversaries.

**Extended analyses.** More analyses are in Appendix G, including task-vector re-weighting ablations, hyperparameter analyses, and cost comparisons, all of which corroborate the efficacy of PISTOLE.

## 5 CONCLUSION

Motivated by our gradient-similarity analyses between natural and robust VLMs, we introduced PISTOLE, a prediction stability–aware task-vector merging framework that composes off-the-shelf natural and robust VLMs without retraining. PISTOLE forms complementary, gradient-informed masks and refines them along adversarial parameter trajectories, weighting steps by a curvature-linked prediction criticality index. Our theorems bound cross-objective interference and show that this index tracks Hessian trace, explaining why the merge gravitates toward flatter, more generalizable regions. Empirically, PISTOLE consistently improves the clean-robust trade-off across 14 datasets and scenarios, and transfers as a drop-in encoder to downstream tasks. Rigorous ablations and curvature analyses validate each component and quantify its impact on the clean-robust frontier.

## ACKNOWLEDGMENT

This research/project is partly supported by the National Research Foundation, Singapore under its AI Singapore Programme (AISG Award No: AISG3-RP-2022-031), partly by the MTI under its AI Centre of Excellence for Manufacturing (AIMfg) (Award W25MCMF014), and the College of Computing and Data Science, Nanyang Technological University.

## ETHICS STATEMENT

This work studies model robustness and parameter-space merging for vision-language models. Our experiments use publicly available datasets and model checkpoints under their original licenses. No personally identifiable or sensitive data is introduced. We evaluate robustness with standard adversarial attacks to stress-test models in a defensive setting. As with most robustness research, there is potential dual use: insights that improve defenses could also inform stronger attacks. To mitigate misuse, our paper reports evaluations and ablations strictly for benchmarking and does not target real users or deployed systems. We encourage responsible release and deployment practices, including clearly labeling merged checkpoints, documenting training/merging procedures, and re-validating safety filters when models are adapted. From an environmental perspective, parameter-space merging substantially reduces computational costs compared with repeated adversarial fine-tuning. We are not aware of disparate-impact risks unique to our method beyond those inherited from the underlying datasets and models. Nonetheless, we recommend auditing downstream applications for distribution shift and fairness where appropriate.

## REPRODUCIBILITY STATEMENT

We organize the paper and appendix to enable end-to-end replication using publicly available models and datasets. Complete experimental configurations, including datasets, preprocessing, evaluation splits, and attack settings, are centralized in Appendix C. Dataset coverage and zero-shot evaluation suites are detailed in Appendix C.1. The task-vector construction and merging/evaluation protocol (backbones, default hyperparameters, and attack parameters) appear in Appendix C.2, while downstream transfer setups for image captioning, VQA, object hallucination, and ScienceQA CoT are specified in Appendix C.3. Theoretical assumptions and complete proofs are provided in Appendix D. Hyperparameter choices, sensitivity studies, and search ranges are summarized in Appendix G.2. Tables and figures in the main text refer back to these sections so that all reported clean and robust accuracies can be reproduced under the stated configurations. We stress that our code and checkpoints will be publicly available to facilitate independent verification and reuse.

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

## A  APPENDIX SUMMARY

This appendix provides background on adversarial fine-tuning for VLMs (Section B), followed by full experimental configurations (Section C). We then present the complete theoretical analyses (see Section D) with the algorithmic specification of our **PISTOLE** method. We supply qualitative visualizations of zero-shot transfer under clean and adversarial inputs (Section F). Extended studies report ablations on re-weighting, hyperparameter sensitivity, computational cost comparisons, and also the performance comparison with standard task vector merging approaches are listed in Section G. Finally, we conclude with limitations and broader impact of our work in Section H with an LLM usage declaration in Section I.

## B  DIVERSE ADVERSARIAL FINE-TUNING SCHEMES

Below, we provide details regarding adversarial fine-tuning schemes in the context of Vision-Language Models (VLMs) for a more comprehensive background introduction.

**TeCoA (Mao et al., 2023) [In-Distribution Robustness].** Given image-text pairs $\mathcal{D}$, standard adversarial fine-tuning (TeCoA) (Mao et al., 2023) is framed as a minimax optimization to improve adversarial robustness of the CLIP models:

$$\min_{\boldsymbol{\theta}_{\mathrm{I}}} \mathbb{E}_{(\mathbf{x},c)\sim\mathcal{D}} \left[ \max_{\|\boldsymbol{\delta}\|_\infty \leq \epsilon} \mathcal{L}_{\mathrm{CE}}\big(\mathbf{p}(\mathbf{x}+\boldsymbol{\delta}), \mathbf{y}(c)\big) \right], \tag{13}$$

**PMG (Wang et al., 2024) [OOD Robustness].** Motivated by the inherent over-fitting of TeCoA (Mao et al., 2023) with generalization degradation, *PMG* (Wang et al., 2024) leveraged the prediction-level guidance from the vanilla pre-trained VLM with a regularization of clean samples for the target model to conduct adversarial fine-tuning, as follows:

$$\min_{\boldsymbol{\theta}_{\mathrm{I}}} \mathbb{E}_{(\mathbf{x},c)\sim\mathcal{D}} \Big[ \max_{\|\boldsymbol{\delta}\|_\infty \leq \epsilon} \mathcal{L}_{\mathrm{CE}}\big(\mathbf{p}(\mathbf{x}+\boldsymbol{\delta}), \mathbf{y}(c)\big)$$
$$+ \lambda_1 \cdot \mathcal{L}_{\mathrm{KL}}(\mathbf{p}_{\mathrm{orig}}(\mathbf{x})\|\mathbf{p}(\mathbf{x}+\boldsymbol{\delta})) + \lambda_2 \cdot \mathcal{L}_{\mathrm{KL}}(\mathbf{p}(\mathbf{x})\|\mathbf{p}(\mathbf{x}+\boldsymbol{\delta})) \Big], \tag{14}$$

where $\mathcal{L}_{\mathrm{KL}}$ represents the Kullback–Leibler divergence to align predictions, and $\mathbf{p}_{\mathrm{orig}}$ denotes the prediction of the vanilla pre-trained CLIP model (Radford et al., 2021). $\lambda_1$ and $\lambda_2$ are the corresponding loss weighting factors.

**FARE (Schlarmann et al., 2024) [OOD Robustness].** To enhance the robustness generalization capability across diverse vision-language tasks, Schlarmann *et al.* (Schlarmann et al., 2024) proposed an unsupervised adversarial fine-tuning approach, dubbed *FARE*, to adversarially optimize feature-level discrepancies in an unsupervised scheme:

$$\min_{\boldsymbol{\theta}_{\mathrm{I}}} \mathbb{E}_{(\mathbf{x},c)\sim\mathcal{D}} \left[ \max_{\|\boldsymbol{\delta}\|_\infty \leq \epsilon} \left\| \mathbf{F}_{\mathrm{orig}}(\mathbf{x}) - \mathbf{F}(\mathbf{x}+\boldsymbol{\delta}) \right\|_2^2 \right], \tag{15}$$

where $\mathbf{F}(\cdot)$ denotes the image encoder of the CLIP model for fine-tuning, while $\mathbf{F}_{\mathrm{orig}}(\cdot)$ is the image encoder of the vanilla pre-trained CLIP model as the frozen reference.

**TGA (Yu et al., 2024) [In-Distribution Robustness].** Built on TeCoA (Mao et al., 2023), Text-Guided Attention (TGA) adversarially fine-tunes the image encoder while aligning text-guided attention maps, computed by correlating per-patch visual tokens with a frozen text embedding of the class prompt (from the original CLIP). It (i) pulls the target model's attention on adversarial images toward the original model's clean-image attention and (ii) keeps the target's clean-image attention close to the original, aiming to boost robustness with eroding marginal clean accuracy.

## C  FULL EXPERIMENTAL CONFIGURATIONS

This section details datasets used, task vector merging/evaluation settings, and downstream transfer protocols used in our experiments and analyses with our *PISTOLE* method.

## C.1 DATASET DESCRIPTION

Following prior work on robust VLMs (Mao et al., 2023; Schlarmann et al., 2024; Dong et al., 2025d;a), we train on the ImageNet-1k training split (Deng et al., 2009) and report classification on the *validation* split (test labels are unavailable). Zero-shot classification is further evaluated on 13 datasets covering:

- **Natural objects:** STL-10 (Coates et al., 2011), CIFAR-10/100 (Krizhevsky et al., 2009), Caltech-101 (Fei-Fei et al., 2004).
- **Fine-grained:** Stanford Cars (Krause et al., 2013), Oxford-IIIT Pets (Parkhi et al., 2012), Flowers-102 (Nilsback & Zisserman, 2008), FGVC-Aircraft (Maji et al., 2013).
- **Textures:** DTD (Cimpoi et al., 2014).
- **Remote sensing:** EuroSAT (Helber et al., 2019).
- **Medical:** PCAM (Veeling et al., 2018).
- **Robustness variants:** ImageNet-R (Hendrycks et al., 2021) and ImageNet-S (Wang et al., 2019).

We also assess zero-shot transfer on a series of downstream image–text understanding tasks: image captioning on COCO (Lin et al., 2014) and Flickr30k (Plummer et al., 2015); Visual Question Answering (VQA) on TextVQA (Singh et al., 2019), VQAv2 (Goyal et al., 2017), and VizWiz (Gurari et al., 2018); object hallucination evaluation via POPE (Li et al., 2023); and multimodal Chain-of-Thought (CoT) reasoning on ScienceQA dataset (Lu et al., 2022).

## C.2 IMPLEMENTATION DETAILS (ZERO-SHOT IMAGE CLASSIFICATION)

**Default setup.** Unless noted, we use CLIP (Radford et al., 2021) with a ViT-L/14 image encoder. In line with standard task arithmetics (Ilharco et al., 2023), all the task vectors are the parameter differences of the vision encoder between fine-tuned and pre-trained CLIP models. The natural VLM is trained by Empirical Risk Minimization (ERM) on clean data, while the robust VLM follows PMG (Wang et al., 2024). Adversarial examples are generated by 10-step PGD (Madry et al., 2018) at $\ell_\infty$ radius $\epsilon = 2/255$ and step size $\alpha = 1/255$. We form task vectors ($\tau_{\text{nat}}, \tau_{\text{rob}}$) and merge them via **PISTOLE** with default mixing factor $\lambda = 0.2$ (Eq. (11)) and the mask sharpness factor $\kappa = 2$ (Eq. (6)). Following adversarial weight training in single-modal architectures (Wu et al., 2020; Dong et al., 2024a), we adopt the adversarial parameter radius factor $\eta = 1 \times 10^{-3}$ (Eq. (8)). For backbone studies (Table 2), we consider both a large architecture CLIP ViT-H/14 and a lightweight one CLIP ViT-B/14. For fine-tuning the CLIP models, we optimize with AdamW (Loshchilov & Hutter, 2019) (betas $(0.9, 0.95)$), a cosine schedule with linear warm-up to $1 \times 10^{-5}$ over 2 epochs. For PEFT experiments (Table 4), we use the LoRA (Hu et al., 2022) scheme specifically on attention blocks. All experimental runs and empirical analyses use eight NVIDIA H100 GPUs.

**Evaluation protocol.** Following prior adversarial learning works (Mao et al., 2023; Schlarmann et al., 2024; Dong et al., 2024c;b), we report clean accuracy and robust accuracy under AutoAttack (AA) (Croce & Hein, 2020) with $\epsilon = 2/255$ unless specified. Note that AA is an ensemble adversarial attack of diverse scenarios for practical reliability assessment. All the robustness evaluation results are based on adaptive attacks for a fair comparison. Zero-shot classification follows the standard CLIP inference/evaluation protocol: cosine similarity between image features and class-prompt text features, selecting the highest-scoring prediction index as the result.

## C.3 DETAILS OF DOWNSTREAM TASK EXTENSIONS

For downstream vision-language task evaluations, we replace the ViT-L/14 vision encoder in LLaVA-1.5-7B (Liu et al., 2024) and OpenFlamingo-9B (Awadalla et al., 2023) with our merged encoder with a better clean-robust trade-off. Note that all other components remain fixed. Below, we provide detailed explanations about each vision-language task and its corresponding setup.

**Downstream task extension to image captioning.** We evaluate the CIDEr score (Vedantam et al., 2015) on COCO (Lin et al., 2014) and Flickr30k (Plummer et al., 2015) using LLaVA-1.5-7B and OpenFlamingo-9B with our merged vision encoder swapped in. Adversarial examples are generated with APGD (Croce & Hein, 2020) under an $\ell_\infty$ perturbation budget of $\epsilon = 2/255$, running 100 steps

*per image–reference pair* following (Schlarmann et al., 2024). After each step, we recompute CIDEr and early-stop that example if the score falls below 10 (COCO) or 2 (Flickr30k). For reporting, we apply the adversarial perturbation that achieved the lowest CIDEr score across references.

**Downstream task extension to visual question answering.** For evaluations, we report standard Visual Question Answering (VQA) accuracy (Antol et al., 2015), selecting the five most frequent answers among the ten annotations for each example. Adversarial inputs are crafted with APGD (Croce & Hein, 2020) at the radius of $\epsilon = 2/255$, using targeted perturbations steered by prompt strings (e.g., ``Maybe'', ``Word'') as in (Schlarmann & Hein, 2023; Schlarmann et al., 2024). Model decoding and text prompts follow each framework's default setup unless stated otherwise.

**Downstream task extension to object hallucination (POPE evaluation benchmark).** We assess object hallucination with POPE benchmark (Li et al., 2023) by issuing binary (yes/no) queries about candidate objects under three standard sampling strategies: *Random* (uniform over absent objects), *Popular* (top-$k$ most frequent absent objects), and *Adversarial* (top-$k$ absent objects with highest co-occurrence with present objects). Following (Li et al., 2023), we report the F1 score and also average the score across sampling strategies for a single summary metric.

**Downstream task extension to Science question answering with chain-of-thought.** We evaluate on ScienceQA (Lu et al., 2022) using LLaVA-1.5-7B (Liu et al., 2024) with our PISTOLE-merged ViT-L/14 vision encoder swapped in and all other weights frozen. Each instance provides an image, a question, and four options $\{A, B, C, D\}$. We adopt a chain-of-thought protocol: the model is first prompted to "think step by step" to produce a free-form rationale, then a short follow-up prompt elicits a single final option token. Inputs follow LLaVA defaults (single image and single-turn dialogue). Decoding uses the temperature $T = \{0.0, 0.1, 0.2\}$. Note that the last explicit token among A/B/C/D is taken as the prediction, with fallback to option–text matching if needed. We report accuracy at each $T$ and the mean across temperatures (Table 7).

# D FULL THEORETICAL ANALYSES

## D.1 PROOF OF THEOREM 1

In this section, we provide the complete proof of Theorem 1, establishing first-order attenuation of cross-objective interference under layerwise complementary masking and its monotone dependence on the mask hyperparameters.

**Theorem 3** (Theorem 1 from the main text). *Let* $\mathbf{g}_{nat}^0 := \nabla_{\boldsymbol{\theta}} \mathcal{L}_{nat}(\boldsymbol{\theta})\big|_{\boldsymbol{\theta}=\boldsymbol{\theta}_0}$ *and* $\mathbf{g}_{rob}^0 := \nabla_{\boldsymbol{\theta}} \mathcal{L}_{rob}(\boldsymbol{\theta})\big|_{\boldsymbol{\theta}=\boldsymbol{\theta}_0}$. *Let* $\mathbf{M}_{nat}^{pre} := \left(\mathbf{1} - \tilde{\mathbf{g}}_{rob}\right)^\kappa$ *and* $\mathbf{M}_{rob}^{pre} := \left(\mathbf{1} - \tilde{\mathbf{g}}_{nat}\right)^\kappa$ *be the uncapped complementary masks from Eq. (6), with* $\kappa \geq 1$ *and* $\tilde{\mathbf{g}}_s \in [0,1]^d$ *defined by Eq. (5). For each layer* $l$, *let* $t_{q,l}^{rob}$ *be the q-quantile of* $(\mathbf{M}_{nat}^{pre})_l$ *and* $t_{q,l}^{nat}$ *the q-quantile of* $(\mathbf{M}_{rob}^{pre})_l$ *(empirical quantiles on layer entries).[2] Define the capped masks layerwise by* $(\mathbf{M}_{nat})_l := \min\left((\mathbf{M}_{nat}^{pre})_l, \, t_{q,l}^{rob}\mathbf{1}\right)$ *and* $(\mathbf{M}_{rob})_l := \min\left((\mathbf{M}_{rob}^{pre})_l, \, t_{q,l}^{nat}\mathbf{1}\right)$. *Set* $\rho_{nat} := \max_l t_{q,l}^{nat}$ *and* $\rho_{rob} := \max_l t_{q,l}^{rob}$. *Then for any* $\boldsymbol{\delta} \in \mathbb{R}^d$,

$$\left|\langle \mathbf{g}_{nat}^0, \mathbf{M}_{rob} \odot \boldsymbol{\delta} \rangle\right| \leq \rho_{nat} \|\mathbf{g}_{nat}^0\|_2 \|\boldsymbol{\delta}\|_2, \qquad \left|\langle \mathbf{g}_{rob}^0, \mathbf{M}_{nat} \odot \boldsymbol{\delta} \rangle\right| \leq \rho_{rob} \|\mathbf{g}_{rob}^0\|_2 \|\boldsymbol{\delta}\|_2. \quad (16)$$

*Moreover, if* $\kappa$ *is increased (i.e., sharpening* $(1 - \tilde{\mathbf{g}})^\kappa$*) or any of the layerwise caps* $t_{q,l}^{nat}, t_{q,l}^{rob}$ *are decreased, the right-hand sides in Eq. (16) are monotone nonincreasing.*

*Proof.* We here prove the first bound, while the second is identical with roles swapped.

**Layerwise $\ell_\infty$ control of the capped mask.** According to the definition of the capping, for every layer $l$ and every index $i$ in that layer's index set $\mathcal{I}_l$,

$$0 \leq (\mathbf{M}_{rob})_i \leq t_{q,l}^{nat}.$$

Consequently, if we write $\mathbf{m} := \mathbf{M}_{rob}$ and use the layer partition $\{\mathcal{I}_l\}$, then

$$\|\mathbf{m}\|_\infty = \max_i |m_i| = \max_l \max_{i \in \mathcal{I}_l} m_i \leq \max_l t_{q,l}^{nat} = \rho_{nat}. \quad (17)$$

---

[2] Any standard definition of the empirical q-quantile with $q \in (0, 1]$ suffices. We here consider that quantiles are monotone under component-wise decreases.

**Bounding the masked displacement in $\ell_2$.** For any vector $\boldsymbol{\delta}$,

$$\|\mathbf{m} \odot \boldsymbol{\delta}\|_2^2 \;=\; \sum_i m_i^2\, \delta_i^2 \;\leq\; \|\mathbf{m}\|_\infty^2 \sum_i \delta_i^2 \;=\; \|\mathbf{m}\|_\infty^2\, \|\boldsymbol{\delta}\|_2^2.$$

Taking square roots and invoking Eq. (17) yields

$$\|\mathbf{M}_{\text{rob}} \odot \boldsymbol{\delta}\|_2 \;\leq\; \|\mathbf{M}_{\text{rob}}\|_\infty\, \|\boldsymbol{\delta}\|_2 \;\leq\; \rho_{\text{nat}}\, \|\boldsymbol{\delta}\|_2. \tag{18}$$

**First-order inner-product control.** By Cauchy–Schwarz,

$$\left|\langle \mathbf{g}_{\text{nat}}^0,\, \mathbf{M}_{\text{rob}} \odot \boldsymbol{\delta}\rangle\right| \;\leq\; \|\mathbf{g}_{\text{nat}}^0\|_2\, \|\mathbf{M}_{\text{rob}} \odot \boldsymbol{\delta}\|_2.$$

Combining with Eq. (18) gives

$$\left|\langle \mathbf{g}_{\text{nat}}^0,\, \mathbf{M}_{\text{rob}} \odot \boldsymbol{\delta}\rangle\right| \;\leq\; \rho_{\text{nat}}\, \|\mathbf{g}_{\text{nat}}^0\|_2\, \|\boldsymbol{\delta}\|_2,$$

which is the first inequality in Eq. (16).

**Monotonicity in $\kappa$ and caps.** Consider the pointwise map $\phi_\kappa(u) = (1-u)^\kappa$ on $u \in [0,1]$. Since $\kappa \mapsto \phi_\kappa(u)$ is nonincreasing for every fixed $u \in [0,1]$, increasing $\kappa$ makes the pre-cap masks $\mathbf{M}_{\text{rob}}^{\text{pre}}$ and $\mathbf{M}_{\text{nat}}^{\text{pre}}$ component-wise *no larger*. Quantiles are monotone under component-wise decreases: if $\mathbf{a} \leq \mathbf{b}$ element-wise then the empirical $q$-quantile of $\mathbf{a}$ is $\leq$ that of $\mathbf{b}$.[3] Therefore $t_{q,l}^{\text{nat}}$ and $t_{q,l}^{\text{rob}}$ are nonincreasing in $\kappa$. The capping operation ($\mathbf{v} \mapsto \min(\mathbf{v}, t\mathbf{1})$) is also monotone nonexpansive (it cannot increase any coordinate). Thus both masks after capping are nonincreasing in $\kappa$, and so are $\rho_{\text{nat}} = \max_l t_{q,l}^{\text{nat}}$ and $\rho_{\text{rob}} = \max_l t_{q,l}^{\text{rob}}$. Finally, explicitly decreasing any $t_{q,l}^{\text{nat}}$ or $t_{q,l}^{\text{rob}}$ further shrinks the corresponding mask entries and hence cannot increase the right-hand sides in Eq. (16). $\qquad\square$

### D.2 Proof of Theorem 2

**Theorem 4** ([Theorem 2 from the main text]**.** *] Let* PCI *be defined as in Definition 1, where* $\mathbf{p}_c(\mathbf{x};\boldsymbol{\theta}) > 0$ *is twice continuously differentiable in a neighborhood of* $\boldsymbol{\theta}$*. Given* $\mathbf{H}_c(\boldsymbol{\theta}) := \nabla_{\boldsymbol{\theta}}^2 \mathbf{p}_c(\mathbf{x};\boldsymbol{\theta})$*, for sufficiently small* $\eta$*, we have the following approximation:*

$$\text{PCI}(\mathbf{x}, c, \boldsymbol{\theta}) = \frac{\sigma^2}{2}\, \frac{\text{Tr}(\mathbf{H}_c(\boldsymbol{\theta}))}{\mathbf{p}_c(\mathbf{x};\boldsymbol{\theta})} + \mathcal{O}(\sigma^3). \tag{19}$$

*Proof.* Throughout the proof, we take all the expectations with respect to the isotropic and zero-mean perturbation $\boldsymbol{\Delta} \in \mathcal{V}_{\boldsymbol{\theta}}$ at the parameter level introduced in Definition 1. The covariance of $\boldsymbol{\Delta}$ is $\sigma^2 \mathbf{I}_d$, where the per-coordinate second moment is $\sigma^2 := \frac{1}{d}\mathbb{E}_{\boldsymbol{\Delta} \in \mathcal{V}_{\boldsymbol{\theta}}}[\|\boldsymbol{\Delta}\|_2^2] = \frac{\eta^2 \|\boldsymbol{\theta}\|_F^2}{d}$.

Given the mapping $\boldsymbol{\vartheta} \mapsto \mathbf{p}_c(\mathbf{x};\boldsymbol{\vartheta})$ is $C^2$ by definition. The second-order multivariate Taylor's expansion around $\boldsymbol{\theta}$ with Lagrange form remainder yields:

$$\mathbf{p}_c(\mathbf{x};\boldsymbol{\theta}+\boldsymbol{\Delta}) = \mathbf{p}_c(\mathbf{x};\boldsymbol{\theta}) + \underbrace{\nabla_{\boldsymbol{\theta}}\mathbf{p}_c^\top(\mathbf{x};\boldsymbol{\theta})\,\boldsymbol{\Delta}}_{(a)}$$
$$+ \underbrace{\tfrac{1}{2}\boldsymbol{\Delta}^\top \mathbf{H}_c(\boldsymbol{\theta})\boldsymbol{\Delta}}_{(b)} + \mathcal{O}(\|\boldsymbol{\Delta}\|^3). \tag{20}$$

Recall that isotropy and zero mean imply $\mathbb{E}[\boldsymbol{\Delta}] = \mathbf{0}$. Thus, the first-order term (a) vanishes under expectation $\mathbb{E}[\nabla_{\boldsymbol{\theta}}\mathbf{p}_c^\top(\mathbf{x};\boldsymbol{\theta})\,\boldsymbol{\Delta}] = 0$. For the quadratic term (b), we apply $\mathbb{E}[\boldsymbol{\Delta}^\top \mathbf{H}_c(\boldsymbol{\theta})\boldsymbol{\Delta}] = \text{Tr}(\mathbf{H}_c(\boldsymbol{\theta})\mathbb{E}[\boldsymbol{\Delta}\boldsymbol{\Delta}^\top]) = \sigma^2\,\text{Tr}(\mathbf{H}_c(\boldsymbol{\theta}))$ for isotropic covariance. Hence, we obtain:

$$\mathbb{E}_{\boldsymbol{\Delta}}[\mathbf{p}_c(\mathbf{x};\boldsymbol{\theta}+\boldsymbol{\Delta})] = \mathbf{p}_c(\mathbf{x};\boldsymbol{\theta}) + \frac{\sigma^2}{2}\,\text{Tr}(\mathbf{H}_c(\boldsymbol{\theta})) + \mathcal{O}(\sigma^3). \tag{21}$$

Taylor's theorem bounds the truncation error in Eq. (20) by $\mathcal{O}(\|\boldsymbol{\Delta}\|_2^3)$. Under the isotropic law in $\mathcal{V}_{\boldsymbol{\theta}}$, we have $\mathbb{E}\|\boldsymbol{\Delta}\|_2^2 = d\sigma^2$, so Hölder's inequality with exponents $(3/2, 3)$ gives:

$$\mathbb{E}\|\boldsymbol{\Delta}\|_2^3 \leq (\mathbb{E}\|\boldsymbol{\Delta}\|_2^2)^{3/2} = (d\sigma^2)^{3/2} = d^{3/2}\,\sigma^3. \tag{22}$$

---

[3]Formally, for any $t$, the empirical CDFs satisfy $F_{\mathbf{a}}(t) \geq F_{\mathbf{b}}(t)$, hence $\inf\{t : F_{\mathbf{a}}(t) \geq q\} \leq \inf\{t : F_{\mathbf{b}}(t) \geq q\}$.

---

**Algorithm 1 PredIction STability-aware mOdeL mErging (PISTOLE)**

---

**Input**: natural and robust CLIP models $(f_{\boldsymbol{\theta}_{\text{nat}}}, f_{\boldsymbol{\theta}_{\text{rob}}})$ and their task vectors $(\boldsymbol{\tau}_{\text{nat}}, \boldsymbol{\tau}_{\text{rob}})$; dataset $\mathcal{D} = \{(\mathbf{x}, c)\}$; input-PGD steps $m$ and step size $\alpha$; parameter-trajectory steps $K$ and step size $\beta$; parameter-ball radius factor $\eta$; mask temperature $\gamma$ and sharpness $\kappa$; small $\epsilon > 0$; trade-off $\lambda$; quantile cap $q$.

1: **Initialize** accumulated gradients: $\mathbf{G}_{\text{nat}} \leftarrow \mathbf{0}, \mathbf{G}_{\text{rob}} \leftarrow \mathbf{0}$
2: **while** not at the end of task vector merging **do**
3:      Sample $(\mathbf{x}, c) \sim \mathcal{D}$ and set $\hat{\mathbf{x}}^{(0)} \leftarrow \mathbf{x} + 0.001 \cdot \mathcal{N}(\mathbf{0}, \mathbf{I})$
4:      **for** $t = 1, \dots, m$ **do**                 $\triangleright$ PGD to obtain adversarial input for robust branch
5:          $\hat{\mathbf{x}}^{(t)} \leftarrow \hat{\mathbf{x}}^{(t-1)} + \alpha \cdot \text{sign}\big(\nabla_{\hat{\mathbf{x}}^{(t-1)}} \mathcal{L}_{\text{CE}}(\mathbf{p}_{\boldsymbol{\theta}_{\text{rob}}}(\hat{\mathbf{x}}^{(t-1)}), \mathbf{e}(c))\big)$
6:          $\hat{\mathbf{x}}^{(t)} \leftarrow \Pi_{\mathbb{B}(\mathbf{x}, \epsilon)}\big(\hat{\mathbf{x}}^{(t)}\big)$
7:      **end for**
8:      Set $\hat{\mathbf{x}} \leftarrow \hat{\mathbf{x}}^{(m)}, \mathbf{x}_{\text{nat}} \leftarrow \mathbf{x}, \mathbf{x}_{\text{rob}} \leftarrow \hat{\mathbf{x}}$
9:      **for** $s \in \{\text{nat}, \text{rob}\}$ **do**               $\triangleright$ Adversarial parameter trajectories
10:          $\boldsymbol{\theta}_s^{(0)} \leftarrow \boldsymbol{\theta}_s$; $\mathcal{V}_{\boldsymbol{\theta}_s} \leftarrow \{\boldsymbol{\Delta} : \|\boldsymbol{\Delta}\|_F \leq \eta\|\boldsymbol{\theta}_s\|_F\}$
11:          **for** $i = 0, \dots, K - 1$ **do**
12:              $\mathbf{g} \leftarrow \nabla_{\boldsymbol{\theta}} \mathcal{L}_s\big(\mathbf{p}_{\boldsymbol{\theta}}(\mathbf{x}_s), \mathbf{e}(c)\big)\big|_{\boldsymbol{\theta} = \boldsymbol{\theta}_s^{(i)}}$
13:              $\mathbf{u} \leftarrow \mathbf{g}/(\|\mathbf{g}\|_F + \epsilon)$
14:              $\boldsymbol{\theta}_s^{(i+1)} \leftarrow \Pi_{\boldsymbol{\theta}_s + \mathcal{V}_{\boldsymbol{\theta}_s}}\big(\boldsymbol{\theta}_s^{(i)} + \beta\, \mathbf{u}\big)$
15:              $\mathbf{G}_s \leftarrow \mathbf{G}_s + \text{PCI}(\mathbf{x}_s, c, \boldsymbol{\theta}_s^{(i)}) \cdot \mathbf{g}$
16:          **end for**
17:      **end for**
18: **end while**
19:    Per-layer normalization and path-refined masks:
     $\tilde{\mathbf{g}}_s^{\text{path}} \leftarrow \text{Norm}(|\mathbf{G}_s|)^{\gamma}$ for $s \in \{\text{nat}, \text{rob}\}$;    $\mathbf{M}_{\text{nat}}^{\text{path}} \leftarrow (1 - \tilde{\mathbf{g}}_{\text{rob}}^{\text{path}})^{\kappa}, \mathbf{M}_{\text{rob}}^{\text{path}} \leftarrow (1 - \tilde{\mathbf{g}}_{\text{nat}}^{\text{path}})^{\kappa}$
20: **if** quantile cap $q$ is specified **then**
21:    Apply per-layer caps:
     $\mathbf{M}_{\text{nat},\ell}^{\text{path}} \leftarrow \min\big(\mathbf{M}_{\text{nat},\ell}^{\text{path}}, t_{q,\ell}^{\text{rob}}\big), \mathbf{M}_{\text{rob},\ell}^{\text{path}} \leftarrow \min\big(\mathbf{M}_{\text{rob},\ell}^{\text{path}}, t_{q,\ell}^{\text{nat}}\big)$
22: **end if**
23:    Re-weight task vectors and merge:
     $\boldsymbol{\tau}_{\text{nat}}^* \leftarrow \mathbf{M}_{\text{nat}}^{\text{path}} \odot \boldsymbol{\tau}_{\text{nat}}, \boldsymbol{\tau}_{\text{rob}}^* \leftarrow \mathbf{M}_{\text{rob}}^{\text{path}} \odot \boldsymbol{\tau}_{\text{rob}}, \boldsymbol{\tau}^* \leftarrow \lambda\, \boldsymbol{\tau}_{\text{nat}}^* + (1 - \lambda)\, \boldsymbol{\tau}_{\text{rob}}^*$
24: **return** merged vector $\boldsymbol{\tau}^*$ and parameters $\boldsymbol{\theta}_{\text{PISTOLE}}(\lambda) = \boldsymbol{\theta}_0 + \boldsymbol{\tau}^*$

---

Hence, the expectation of the remainder term is $\mathcal{O}(\sigma^3)$, validating the order claimed in Eq. (21).

Subtracting $\mathbf{p}_c(\mathbf{x}; \boldsymbol{\theta})$ from Eq. (21) and dividing by $\mathbf{p}_c(\mathbf{x}; \boldsymbol{\theta}) > 0$ (keeping terms up to $\mathcal{O}(\sigma^2)$) yields:

$$\mathbb{E}\Big[\frac{\mathbf{p}_c(\mathbf{x}; \boldsymbol{\theta} + \boldsymbol{\Delta}) - \mathbf{p}_c(\mathbf{x}; \boldsymbol{\theta})}{\mathbf{p}_c(\mathbf{x}; \boldsymbol{\theta})}\Big] = \frac{\sigma^2}{2} \frac{\text{Tr}(\mathbf{H}_c(\boldsymbol{\theta}))}{\mathbf{p}_c(\mathbf{x}; \boldsymbol{\theta})} + \mathcal{O}(\sigma^3). \tag{23}$$

For sufficiently small $\sigma^2$, the leading term dictates the sign, and the outer absolute value in the definition of our proposed PCI (Definition 1) therefore keeps the magnitude and removes the sign. Consequently, we obtain:

$$\text{PCI}(\mathbf{x}, c, \boldsymbol{\theta}) = \frac{\sigma^2}{2} \frac{\text{Tr}(\mathbf{H}_c(\boldsymbol{\theta}))}{\mathbf{p}_c(\mathbf{x}; \boldsymbol{\theta})} + \mathcal{O}(\sigma^3), \tag{24}$$

establishing the quadratic estimation in Eq. (19). $\qquad\square$

## E    PISTOLE: FULL ALGORITHMIC SPECIFICATION

This section instantiates the procedure described in the main text (*cf.* Sections 3.3 and 3.4). The routine first constructs robust inputs for the `robust` branch via the PGD adversary generation scheme, then traces short *adversarial parameter trajectories* around each fine-tuned solution. Along these

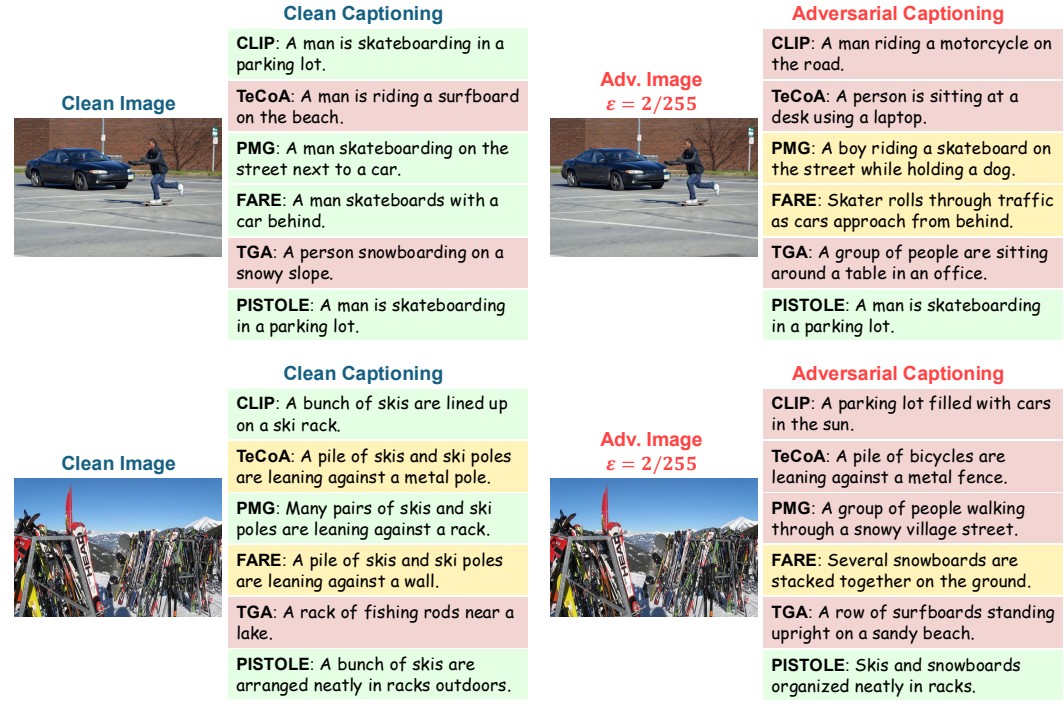

Figure 6: **Image captioning** under clean and adversarial inputs ($\ell_\infty$-norm perturbation $\epsilon = 2/255$) using LLaVA-1.5 coupled with vision encoders from the compared methods on COCO. PISTOLE maintains semantic consistency across perturbations, whereas alternatives often drift or hallucinate.

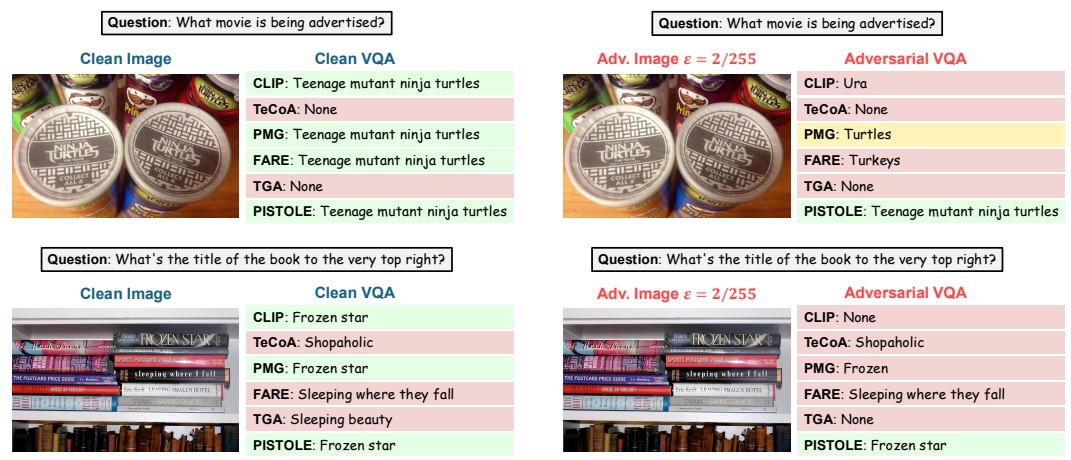

Figure 7: Qualitative **visual question answering** under clean samples and their adversarial counterparts ($\epsilon = 2/255$) on TextVQA, using LLaVA-1.5 with different robust vision encoders. PISTOLE preserves correct answers across both conditions, whereas baselines often drift or abstain.

trajectories, it accumulates *PCI*-weighted gradients, converts them to per-layer normalized sensitivity scores, and forms complementary, path-refined masks. Finally, it re-weights the natural/robust task vectors and merges them with mixing coefficient $\lambda$.

In practice, the method is training-free and efficient: small step counts suffice; $\epsilon$ stabilizes normalization; $(\gamma, \kappa)$ tune mask dynamic range and selectivity; and the quantile cap $q$ enforces a first-order non-interference budget. Unless otherwise stated, we apply our PISTOLE method to the vision encoder parameters, but the specification in Algorithm 1 is architecture-agnostic.

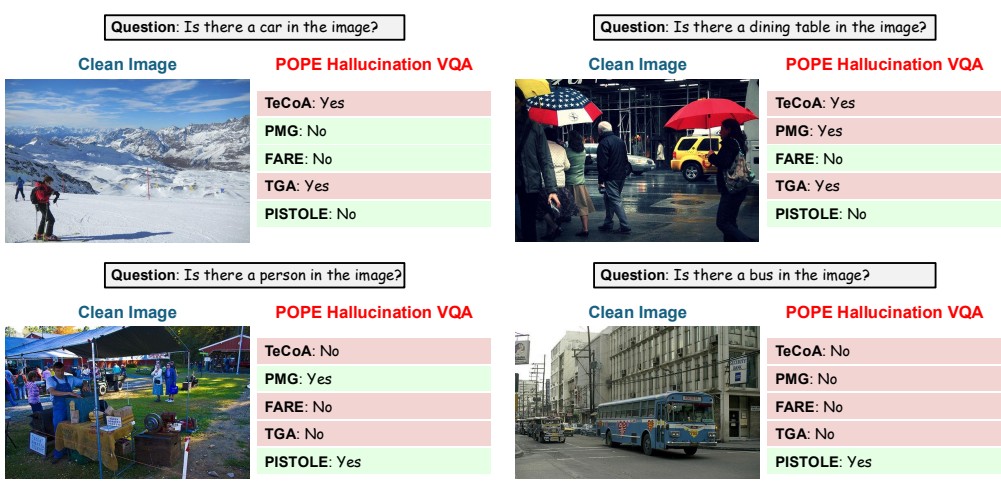

Figure 8: Visual examples on **POPE hallucination benchmark** using LLaVA 1.5 with vision encoders from different adversarial VLM learning schemes. PISTOLE reduces yes/no hallucinations and maintains pixel-grounded responses under perturbation.

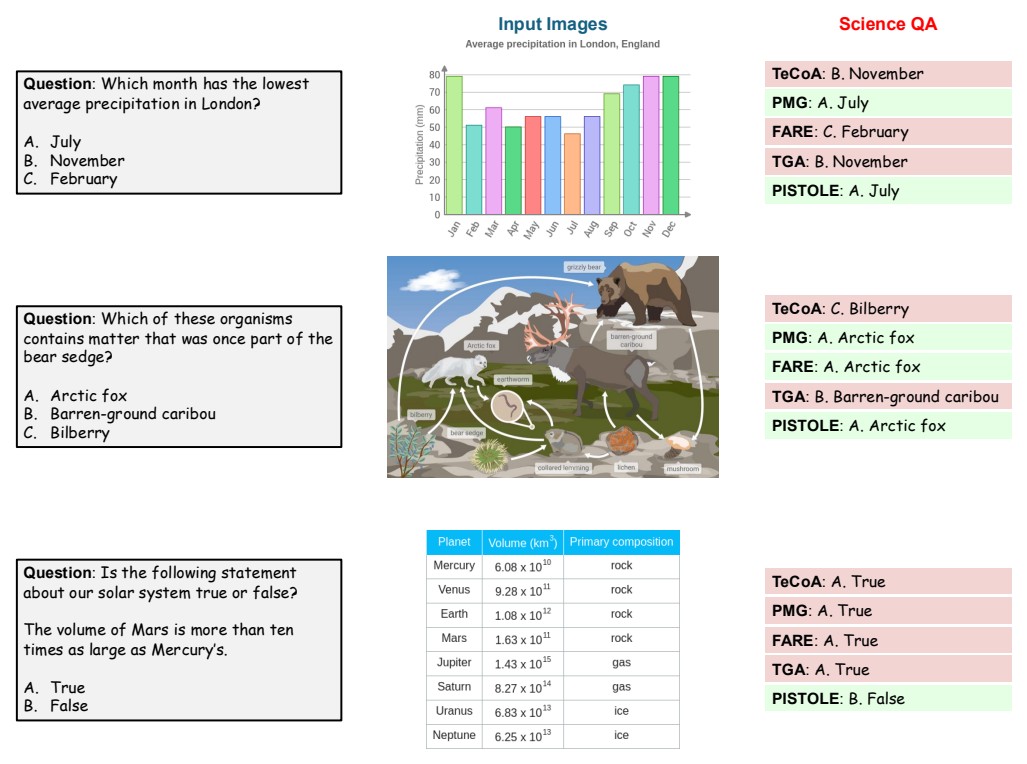

Figure 9: Qualitative examples for **science question answering w/ CoT** using LLaVA-1.5 with different robust vision encoders. Our PISTOLE method maintains evidence-consistent answers under perturbations, while alternatives often drift toward prior-biased choices.

## F  VISUALIZATIONS OF ZERO-SHOT TRANSFER TO DOWNSTREAM TASKS

In this section, we present qualitative comparisons across adversarial learning baselines and our task vector merging method (**PISTOLE**) for zero-shot transfer across diverse downstream vision-language tasks, *e.g.*, captioning and visual question answering, under clean inputs and adversaries.

**Zero-shot transfer to image captioning.** Figure 6 shows that equipping LLaVA-1.5 with PISTOLE's merged encoder yields captions that remain semantically stable from clean to adversarial images ($\ell_\infty$-norm perturbation $\epsilon = 2/255$). Competing encoders frequently drift across domains or

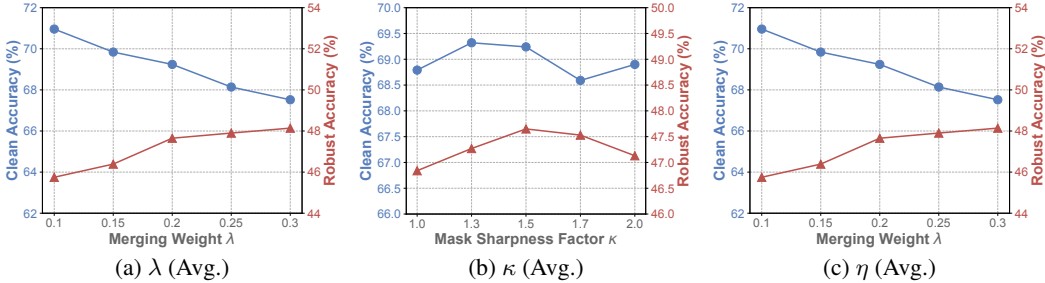

| (a) $\lambda$ (Avg.) | (b) $\kappa$ (Avg.) | (c) $\eta$ (Avg.) |

Figure 10: Hyper-parameter (merge weight $\lambda$, mask sharpness factor $\kappa$, and parameter-trajectory radius scaling factor $\eta$) sensitivity analyses of PISTOLE.

hallucinate objects/attributes under perturbation, while PISTOLE preserves object grounding and scene semantics—evidence of stronger invariance to input attacks and distribution shift.

**Zero-shot transfer to visual question answering.** Figure 7 illustrates two TextVQA cases in which PISTOLE maintains answer consistency from clean to adversarial inputs (*e.g.*, "Teenage Mutant Ninja Turtles" and "Frozen star"), while competing encoders frequently switch to incorrect strings or "None". These qualitative trends align with our quantitative gains associated with a higher sum of clean and robust VQA accuracy shown in the main text.

**Zero-shot transfer to POPE object hallucination.** We here illustrate POPE object hallucination cases across its question-sampling protocols in Figure 8. The baseline encoders frequently follow language priors, thus answering "Yes" for common objects and "No" for unlikely ones, regardless of image evidence. In contrast, our PISTOLE method aligns decisions with the pixels, lowering both false positives (hallucinated objects) and false negatives (missed objects), and preserving consistent judgments between clean and adversarial inputs.

**Zero-shot transfer to science question answering w/ CoT.** Figure 9 contrasts predictions on representative ScienceQA items. Under adversarial perturbations, prior robust CLIP variants frequently select high-prior but visually unsupported options, whereas PISTOLE remains aligned with the chart/diagram context and the underlying facts, yielding stable, evidence-grounded answers. These cases suggest that stabilizing the vision backbone with PISTOLE curbs brittle pattern-matching and supports more reliable multi-step reasoning.

## G  EXTENDED ANALYSES OF PISTOLE

### G.1  EFFECT OF TASK VECTOR RE-WEIGHTING

We compare three re-weighting strategies for task vector merging: i) *no mask* (uniform addition), (ii) a *random mask* matching the layerwise sparsity, and (iii) *gradient-informed stability masks* in Eq. (6) in Table 11. Uniform addition yields the weakest trade-off, reproducing

Table 11: Avg. performance (%) of our PISTOLE with diverse **task vector re-weighting schemes**.

| Re-Weighting Scheme | Clean | Robust | Sum |
|---|---|---|---|
| No Mask | 66.57 | 44.54 | 111.11 |
| Random Mask | 67.32 | 44.95 | 111.27 |
| Gradient-Informed Mask | **69.24** | **47.65** | **116.89** |

the near-linear clean–robust antagonism. Random masking offers small gains by incidentally pruning conflicts but lacks guarantees. In contrast, our gradient-informed strategy consistently achieves the best trade-off, bending the frontier toward interior optima by suppressing counterpart-sensitive coordinates. This aligns with Theorem 1 and Corollary 1, which proves that complementary masks contract cross-objective first-order interference, unlike other schemes.

### G.2  HYPERPARAMETER SENSITIVITY ANALYSES.

In this section, we study the effect of three core hyperparameters in our **PISTOLE**, the merge weight $\lambda$ (Eq. (11)), the mask sharpness factor $\kappa$ (Eq. (6)), and the parameter-trajectory radius scaling factor $\eta$ (Eq. (8)), while holding all other settings fixed. Figures 10 plot both the average clean and robust accuracy across the 14 evaluation datasets.

**Merge weight** $\lambda$. Sweeping $\lambda$ from 0.1 to 0.3 reveals the expected trade-off: clean accuracy increases with larger weight on the natural task vector, while robustness decreases roughly monotonically. The frontier is bowed (not linear), yielding an interior optimum of the Clean+Robust **Sum** near $\lambda = 0.2$, which we adopt as default. Trends are stable across seeds, indicating that $\lambda$ primarily sets the accuracy–robustness operating point rather than introducing instability.

**Mask sharpness factor** $\kappa$. Sharpening the complementary masks by increasing $\kappa$ (Eq. (6)) improves clean accuracy and raises robust accuracy up to an interior peak, after which robustness plateaus or dips slightly. The trade-off is maximized near $\kappa = 1.5$, which we use by default to attenuate counterpart-sensitive coordinates without over-pruning.

**Parameter-trajectory radius scaling factor** $\eta$. We can observe that $\eta$ controls the neighborhood explored by the adversarial parameter trajectory: too small under-explores the loss geometry, while too large drifts off the manifold. As shown in the attached sweep, both clean and robust accuracy peak at a moderate radius $\eta = 1 \times 10^{-3}$. It can also be seen that smaller parameter-level perturbation radii yield limited gains, and larger ones degrade the trade-off.

### G.3    COMPUTATIONAL COST COMPARISONS

**Training-free merging.** PISTOLE operates on *off-the-shelf* naturally and adversarially fine-tuned VLMs and performs a *one-shot* merge: we estimate stability masks from a small calibration split (no epochs of weight updates), then apply element-wise reweighting to the two task vectors and

Table 12: Computational cost (training time) comparison between PISTOLE and other adversarial learning methods.

| Method | Clean | Robust | Sum | Time |
|---|---|---|---|---|
| TeCoA | 61.56 | 43.26 | 104.82 | 6.2 hours |
| PMG | 64.46 | 45.74 | 110.20 | 8.6 hours |
| FARE | 65.50 | 42.97 | 108.47 | 7.6 hours |
| TGA | 62.11 | 45.19 | 107.30 | 8.0 hours |
| **PISTOLE** | **69.24** | **47.65** | **116.89** | **0.8 hours** |

compose the final encoder. In contrast, prior adversarial VLM learning approaches run full optimization loops with inner PGD steps and (often) model forward propagation, incurring substantial GPU time. To make costs comparable across methods, we count gradient evaluations and auxiliary forwards. As shown in Table 12, we can observe that our PISTOLE method attains the best clean–robust trade-off while being ~*8–11× faster* than prior adversarial fine-tuning baselines.

**Complexity analysis.** Let $N_c$ be the size of the (small) calibration split, $m$ the number of PGD steps for adversarial inputs, and $K$ the number of parameter-trajectory steps. PISTOLE does: (i) $m$ input-gradient evaluations to generate adversarial inputs for the robust branch (same inner loop as standard adversarial training), and (ii) for each branch, $K$ parameter-gradient evaluations along the adversarial parameter trajectory, each weighted by PCI. Thus, the total number of gradient evaluations to estimate the masks is $O(N_c(m + 2K))$, run once over a calibration split with no weight updates and no multi-epoch optimization loop. In contrast, adversarial fine-tuning over $E$ epochs on the full training set of size $N$ has complexity $O(EN(m + 1))$, since each iteration both runs PGD and performs a parameter update. In practice, $E$ is large (multiple epochs), while $K$ is a small constant, and $N_c \ll N$, so the overall cost of PISTOLE is substantially lower even though we do multiple forward–backward passes during calibration.

### G.4    COMPARISON WITH STANDARD TASK VECTOR MERGING METHODS

Recall from Section 2 that Ties-Merging (Yadav et al., 2023) enforces sign-consistent sparsification and AdaMerging (Yang et al., 2024) learns per-parameter weights for multi-task settings. Table 13 compares our PISTOLE with these standard parameter-space merging methods that combine the same naturally and robustly fine-tuned CLIP models in the iden-

Table 13: Average performance (%) over 14 datasets for merging the same natural and robust CLIP models under the identical configuration.

| Task Vector Merging | Clean | Robust | Sum |
|---|---|---|---|
| Vanilla Merging | 66.57 | 44.54 | 111.11 |
| Ties-Merging | 67.91 | 46.27 | 114.18 |
| AdaMerging | 68.23 | 46.52 | 114.75 |
| **PISTOLE** | **69.24** | **47.65** | **116.89** |

tical configuration. Across 14 datasets, our method attains the best clean and robust accuracy, improving the clean-robust trade-off over naive task-vector addition, Ties-Merging, and AdaMerging. We attribute these gains to modeling *parameter-space sensitivity and local loss geometry*: PISTOLE

Figure 11: On an adversarially perturbed image, both the robust source model and PISTOLE output a wrong answer, despite the natural model giving a partially correct answer on the clean input, illustrating failures that merging cannot fix.

uses gradient-informed complementary masks and refines them along adversarial parameter trajectories to account for sensitivity and curvature during merging—factors overlooked by prior methods.

### G.5 FURTHER EXPLANATION OF THE "TRAINING-FREE" ASSUMPTION

Our intent is to emphasize that PISTOLE performs no additional gradient-based optimization for each downstream task once such source models are available. First, this assumption is aligned with common practice in the task-vector/model-editing literature Ilharco et al. (2023); Ortiz-Jimenez et al. (2023), where methods start from already fine-tuned checkpoints and apply post-hoc parameter-space operations. In the VLM ecosystem, high-quality natural and robust checkpoints (*e.g.*, CLIP-/OpenCLIP-style models and their adversarially trained/fine-tuned counterparts) are increasingly released and reused as off-the-shelf backbones. PISTOLE is designed precisely for this regime: given pre-existing natural and adversarial models, we can cheaply obtain a continuum of merged models with improved clean–robust trade-offs without any further training.

Second, even when a pair of natural/robust models must be trained once, this one-time cost is amortized over many downstream tasks/domains. In contrast, standard adversarial fine-tuning typically re-optimizes the model for each new target task. While our PISTOLE exhibits generalizable robustness across diverse downstream vision-language tasks.

### G.6 ANALYSIS ON FAILURE CASES

Figure 11 illustrates a typical failure mode that PISTOLE cannot fix. Our method operates by interpolating and masking between a natural and an adversarially trained model in parameter space, so it can only reshuffle how much each endpoint contributes. When both source models systematically fail on certain patterns (*e.g.*, rare classes or heavily shifted domains), the merged model likewise produces wrong answers and may even accumulate errors, as seen in the adversarial example where both the robust source model and PISTOLE are incorrect despite the natural model being partially correct.

### G.7 BROADER PARAMETER-SPACE CONTEXT (MORE DISCUSSION)

Recent studies have made inspiring progress toward trustworthy large language models from different perspectives. Self-Debias Feng et al. (2026) have made inspiring progress in trustworthy large language models, especially in mitigating social biases and enabling intrinsic self-correction. UniFLE Zhao et al. (2026) has achieved promising advances in improving the safety of LLMs, particularly in defending against weight-poisoning backdoor attacks. Beyond these broader efforts on trustworthy and safe foundation models, several recent VLM-specific methods also operate directly in parameter space. WATT Osowiechi et al. (2024) adapts CLIP under domain shift via test-time updates followed by weight averaging of the adapted parameters, improving test-time robustness. GeoLangBind Xiong et al. (2025) trains a remote-sensing VLM and uses a progressive multimodal weight-merging strategy to aggregate knowledge from multiple visual backbones within a single VLM. MoTE Zhu et al. (2024) adds temporal experts on top of a VLM and employs weight-merging regularization in parameter space to enhance the trade-off. Our work is complementary: instead of merging across domains, backbones, or experts, PISTOLE performs stability-aware merging between natural and adversarially trained vision encoders to reconcile clean accuracy and robustness.

# H    LIMITATIONS AND BROADER IMPACT.

## H.1    BROADER IMPACT

PISTOLE targets a central safety concern in foundation VLMs: robustness to adversarial perturbations. By reconciling clean accuracy and robustness through a **training-free**, **plug-and-play** merge of off-the-shelf natural and robust task vectors, our approach can make multimodal systems more **reliable in downstream applications** (captioning, VQA, hallucination mitigation, and scientific QA). The method is compute-efficient, requiring only a short calibration pass and no weight updates, thereby lowering the environmental and financial cost of robustness compared to full adversarial fine-tuning. Because PISTOLE composes existing checkpoints rather than collecting new data, it also **eases reproducibility and facilitates community vetting**. At the same time, improved robustness should be paired with **standard safeguards** (bias audits, red-teaming, and attack-aware evaluation) to ensure equitable performance across subpopulations and **responsible deployment**.

## H.2    LIMITATIONS

While PISTOLE shows strong empirical gains and formal guarantees, several limitations remain. We note them alongside why their impact is limited or how we partially mitigate them.

- **Dependence on off-the-shelf task vectors.** PISTOLE assumes access to natural and robust fine-tuned VLMs to form task vectors. In practice, this is a minor constraint: high-quality CLIP-family checkpoints (natural and adversarial) are widely available in open-source repositories, and our method is agnostic to the specific recipe used to produce them. Moreover, Section 4.3 (Tables 9&10) shows robustness to the choice of source models, and our masks provably attenuate cross-objective interference (Theorem 1).

- **Inheritance of upstream biases.** Merging cannot remove biases present in the component models and may propagate spurious correlations. Our gradient-informed masks down-weight counterpart-sensitive (often brittle) coordinates, which empirically reduces hallucination and improves grounding, but it does not replace fairness auditing. We report object-hallucination and reasoning improvements, while broader bias assessments are a valuable direction for future work.

- **Scope of architectures and tasks.** Most experiments use CLIP-like encoders and open-vocabulary classification/captioning/VQA. Although we show transfer across backbones and tasks, coverage is not exhaustive (e.g., video, speech–vision). The merge is model-agnostic and only requires gradients for calibration, and our curvature results (Figure 5, Theorem 2) suggest applicability beyond the tested settings/applications.

# I    LLM USAGE DECLARATION/DISCLOSURE.

We used a Large Language Model (LLM) (*e.g.*, ChatGPT-5) solely for polishing: grammar, wording, and LaTeX phrasing. **The LLM did not generate ideas, methods, experiments, analyses, or results.** All technical content and claims were authored and verified by us. Outputs were reviewed and edited by the authors, and all citations/equations were checked manually. No proprietary or non-public data is provided to the model.

