# OpenReview forum: "Tug-of-War No More: Harmonizing Accuracy and Robustness in Vision-Language Models via Stability-Aware Task Vector Merging"
_ICLR.cc/2026/Conference — ICLR 2026 Poster_

### Official Review · Reviewer_uKzG · 2025-10-25

**Soundness:** 4
**Presentation:** 3
**Contribution:** 4
**Rating:** 8
**Confidence:** 4

**Summary:**

The paper investigates a training-free way to merge a naturally fine-tuned CLIP and a robustly (adversarially) fine-tuned CLIP using gradient-informed and prediction-stability masks over task vectors. Instead of uniformly adding task vectors (which they show yields an almost linear clean-vs-robust trade-off), the authors (1) estimate per-parameter stability from gradients of the *other* objective, (2) cap unstable coordinates, and (3) refine these masks along *adversarial parameter trajectories* weighted by a Prediction Criticality Index (PCI) that tracks curvature (Hessian trace). Empirically, PISTOLE bends the frontier and improves the sum of clean+robust performance across 14 datasets. Theoretically, the authors show the masks contract cross-objective first-order interference.

**Strengths:**

1. This paper provides an interesting and clear problem formulation and empirical evidence, where vanilla task vector merging fails to improve the trade-off, yet the proposed method effectively addresses this issue.
2. The paper shows clear and logical writing along with a novel methodology.
3. To the best of my knowledge, this is the first paper to address the accuracy-robustness trade-off without costly adversarial fine-tuning, generalizing across tasks.
4. The theoretical justification matches intuition: masks contract first-order interference, while PCI tracks curvature.
5. The strong ablations and analyses demonstrate the SOTA performance of the proposed method associated with improved efficiency.

**Weaknesses:**

1. The proposed method requires access to both natural and robust fine-tuned models. Does it mean that the quality also depends on them?
2. The “Prediction Criticality Index” is well-motivated mathematically, but the intuition appears late (after Eq. 10). An earlier conceptual explanation could help readers.
3. Recent VLM robustness techniques (e.g., prompt ensembling, token-level defenses) should be discussed in the Related Works.

**Questions:**

1. Could the authors briefly clarify the meaning of kappa in Eq. (6)? It would help if the notation were reintroduced when referenced later in the paper.
2. The authors introduce the term Prediction Criticality Index (PCI) in Section 3.4, but the intuition only becomes clear later. Could you add a short intuitive explanation when it’s first mentioned?
3. Sections 3.3 and 3.4 are mathematically dense. If possible, could you add a brief ‘takeaway’ sentence at the end of each subsection summarizing the main intuition?”

---

> ### Author Response · Authors · 2025-11-20
> **Response to Reviewer uKzG**
>
> # Response to Reviewer uKzG
> We thank the reviewer for constructive feedback.
>
> ### **1. (Weakness-1) Dependence on natural/robust fine-tuned models**
>
> We appreciate the reviewer's question and agree that, by design, PISTOLE's performance depends on the quality of the underlying natural and robust source models.
>
> As shown in Tables 9 and 10, when we swap in stronger natural or robust models, PISTOLE inherits their strengths and yields a better clean–robust trade-off. We will clarify in the problem setup that PISTOLE is an *evolving* merging mechanism whose benefits improve as better natural and adversarially fine-tuned VLMs become available.
>
>
>
> ### **2. (Weakness-2) Intuition for the Prediction Criticality Index**
>
> We thank the reviewer for this suggestion and agree that the intuition for PCI should appear earlier.
>
> Conceptually, the **Prediction Criticality Index (PCI)** measures how *fragile* a prediction is to small parameter perturbations:
>
> - High PCI means that small changes in parameters cause large relative changes in the predicted class probability.
> - Low PCI means that the prediction is stable and confidence-saturated, so small parameter perturbations barely affect the output.
>
> In PISTOLE, we use PCI to up-weight gradients from these fragile predictions along the parameter trajectory, so the merge focuses on stabilizing precisely those modes that are sensitive.
>
>
>
> ### **3. (Weakness-3) Add discussion of recent VLM robustness works**
>
> We thank the reviewer for pointing this out and agree that recent prompt-based and token-level robustness techniques for VLMs should be discussed.
>
> In the revision, we will add a short paragraph in Related Work covering:
>
> - **Prompt ensembling / robust prompt learning.** [a] introduces adversarial prompt tuning to learn a single robust text prompt that improves CLIP's robustness while keeping the main model weights fixed. R-TPT [b] performs test-time prompt tuning to strengthen robustness against adversarial attacks without updating backbone parameters.
> - **Token-level defenses.** For token-level robustness, we here briefly introduce a recent work, SafePTR [c], which is a token-level jailbreak defense for multimodal LLMs that prunes harmful tokens and restores benign features.
>
> We will clearly contrast these with our proposed method in the revised version.
>
>
>
> [a] One prompt word is enough to boost adversarial robustness for pre-trained vision-language models (CVPR 2024)
>
> [b] R-TPT: Improving Adversarial Robustness of Vision-Language Models through Test-Time Prompt Tuning (CVPR 2025)
>
> [c] SafePTR: Token-Level Jailbreak Defense in Multimodal LLMs via Prune-then-Restore Mechanism (Arxiv 2025)
>
>
>
> ### **4. (Question-1) Meaning of $\kappa$**
>
> We thank the reviewer for this suggestion.
>
> In our design, $\kappa$ controls the sharpening of the complementary masks: larger $\kappa$ amplifies the contrast between high- and low-sensitivity coordinates (making the mask more "peaky"). In the revision, whenever we refer back to Eq. (6), we will briefly reintroduce $\kappa$ and explicitly remind readers that it tunes mask sharpening.
>
>
>
> ### **5. (Question-2) Intuitive explanation of the Prediction Criticality Index**
>
> We appreciate the reviewer's suggestion.
>
> When PCI is first introduced, we will add a brief intuition: PCI quantifies **how unstable a prediction is**. Concretely, PCI is high when the model‘s confidence for a class changes rapidly under tiny parameter perturbations (typical of points near decision boundaries), and low when the prediction is already very stable and sits in a flat, confidence-saturated zone. In PISTOLE, such high-PCI points are given more influence when aggregating gradients along the parameter trajectory, so the merge preferentially stabilizes precisely those fragile predictions.
>
>
>
> ### **6. (Question-3) "Takeaway" to summarize the intuition of each theorem**
>
> We thank the reviewer for this suggestion.
>
> We will highlight/strengthen the main intuitions in the green tcolorbox remarks following Theorem 1 (Section 3.3) and Theorem 2 (Section 3.4). In the revision, we will make this much clearer and easier to find by:
>
> - We will explicitly label each green box as a "Takeaway", so readers can naturally treat it as a closing summary.
> - We will highlight that Theorem 1 explains that the complementary masks attenuate cross-objective interference by a tunable factor and thus improve the clean–robust performance trade-off. Theorem 2 explains that PCI is large when the prediction is in high-curvature regions, thus up-weighting high-PCI samples to stabilize fragile prediction modes.

---

> ### Comment · Area_Chair_eAwT · 2025-11-25
>
> Dear Reviewer uKzG,
>
> The authors have responded to your reviews. Please review and provide your feedback and responses.
>
> Best,
>
> Your AC

---

### Official Review · Reviewer_VcKu · 2025-10-30

**Soundness:** 3
**Presentation:** 3
**Contribution:** 3
**Rating:** 4
**Confidence:** 3

**Summary:**

This paper introduces PISTOLE (Prediction STability-aware mOdeL mErging), a training-free framework designed to reconcile the long-standing accuracy–robustness trade-off in CLIP.
Instead of costly adversarial fine-tuning, PISTOLE merges two off-the-shelf models—one naturally fine-tuned and one adversarially fine-tuned—directly in parameter space using task vectors.

To avoid the near-linear degradation observed in naïve merging, the authors propose gradient-informed complementary masks and a prediction criticality index (PCI) that approximates curvature and identifies stable parameters for selective fusion.
Theoretically, PISTOLE is shown to reduce the conflict between natural and robust objectives and guide the merged model toward flatter, more stable parameter regions, thereby improving both accuracy and robustness; empirically, it achieves superior clean–robust trade-offs and transfers effectively to downstream tasks such as captioning, VQA, hallucination detection, and reasoning.

**Strengths:**

1.**Motivation is Clear:** The paper is well-motivated, shifting the focus from model retraining to direct parameter merging, which effectively leverages pretrained foundation models to achieve robustness–accuracy balance with minimal cost.

2.**Method is reasonable:** The approach is conceptually sound, integrating gradient-informed and curvature-aware weighting into the model merging framework, and theoretically demonstrating that (i) complementary masks provably contract cross-objective interference, and (ii) PCI quantitatively approximates curvature through the Hessian trace.

3.**Experiment proves the claim:** Experiments are comprehensive, covering zero-shot classification, captioning, VQA, hallucination, and CoT reasoning, and consistently showing improvements across both clean and robust performance metrics.

**Weaknesses:**

1. PISTOLE assumes access to both naturally and adversarially fine-tuned models, which are themselves costly to obtain. Thus, the “training-free” claim is conditional—effective only once such models are available—and may not reduce the overall computational cost in practice.

2. The paper lacks analysis on failure cases. While quantitative results are strong, does this method works all the time with all the cases? It remains unclear under what conditions the proposed method may fail or underperform.

3. The experimental section could be improved by including more comparisons and discussions with similar task-vector-based methods to better contextualize the contribution.

**Questions:**

In your work, PISTOLE assumes that task vectors can be linearly combined in parameter space
(e.g., $\boldsymbol{\tau}_{\text{add}} = \sum_i \boldsymbol{\tau}_i$).
However, in multimodal encoders, task-specific update directions may not be linearly independent,
and direct addition could introduce interference due to the nonlinearity of the representation space.
Could you clarify on that?

Figure 5 shows that PISTOLE reduces loss curvature during model merging.
Could you clarify how this empirical result quantitatively supports Theorem1 and Theorem2?

---

> ### Author Response · Authors · 2025-11-20
> **Response to Reviewer VcKu (Part-1)**
>
> # Response to Reviewer VcKu (Part-1)
> We thank the reviewer for constructive feedback.
>
> ### **1. (Weakness-1) On the "training-free" assumption and cost of natural/adversarial models**
>
> We appreciate the reviewer’s concern and agree that PISTOLE is “training-free” conditional on having both naturally and adversarially fine-tuned models. Our intent is to emphasize that PISTOLE performs no additional gradient-based optimization for each downstream task once such source models are available.
>
> First, this assumption is aligned with common practice in the **task-vector / model-editing** literature [a,b], where methods start from already fine-tuned checkpoints and apply post-hoc parameter-space operations. In the VLM ecosystem, high-quality natural and robust checkpoints (*e.g.*, CLIP-/OpenCLIP-style models and their adversarially trained/fine-tuned counterparts) are **increasingly released and reused as off-the-shelf backbones**. PISTOLE is designed precisely for this regime: given pre-existing natural and adversarial models, we can cheaply obtain a continuum of merged models with improved clean–robust trade-offs without any further training.
>
> Second, even when a pair of natural/robust models must be trained once, this **one-time cost is amortized over many downstream tasks/domains**. In contrast, standard adversarial fine-tuning typically re-optimizes the model for each new target task. Furthermore, PISTOLE exhibits generalizable robustness across diverse downstream vision-language tasks (See Section 4.2).
>
> In the revision, we will clarify this assumption explicitly in the *Problem Setup* and *Introduction*, and more clearly situate our setting alongside prior task-vector works that make the same starting assumption.
>
>
>
> [a] Editing models with task arithmetic (ICLR 2023).
>
> [b] Task arithmetic in the tangent space: Improved editing of pre-trained models (NeurIPS 2023).
>
>
>
> ### **2. (Weakness-2) Lack of failure-case analysis**
>
> We appreciate the reviewer's positive assessment of the quantitative results and agree that it is important to clarify when our method may fail or underperform.
>
> We find the following potential failure modes:
>
> - **Inherited failures from the source models**. Our method operates by interpolating and masking between a natural and an adversarial model in parameter space. If both source models systematically fail on certain patterns (*e.g.*, rare classes or heavily shifted domains), the merged model cannot "fix" these cases and may even **accumulate errors**. We have added qualitative examples in the appendix illustrating such cases where both endpoints mishandle the input, and our method inherits the failure (See Appendix G.6 for more details).
> - **Naive extension to the text encoder.** As we also discuss in our response to Reviewer QXNC (Response to Comment 5), naively applying PISTOLE to both the vision and text encoders can be problematic: it introduces an extra degree of freedom, where both the image and text embedding spaces move, potentially along different robustness directions. This makes it easier to distort the shared semantic space (image and text embeddings drift in misaligned ways), which in turn can hurt alignment-sensitive downstream tasks. Moreover, in many VLMs, the text encoder (or LLM) is already very strong and broadly pre-trained. Modifying it often brings more harm than benefit, whereas the vision encoder typically has more room to adapt to robustness and new domains. We also provide the corresponding results on ImageNet in the table below:
>
> |Scheme|Vision tuned|Text tuned|Clean|Robust|
> |-|-|-|-|-|
> |TeCoA (image only) + Nat FT VLM (image only)|✓|✗|79.23|62.31|
> |TeCoA (image+text) + Nat FT VLM (image+text)|✓|✓|78.35|61.18|
>
> Extending our method in an alignment-aware way to the text side is left as future work.

---

> > ### Comment · Reviewer_VcKu · 2025-11-21
> > **reply to the author**
> >
> > Thanks for the authors' response. My concern has been addressed, and I support accepting this paper.

---

> > > ### Author Response · Authors · 2025-11-22
> > >
> > > Esteemed Reviewer,
> > >
> > > We thank you for engaging with our rebuttal. Rest assured, all suggestions will be incorporated into our paper. In the meantime, if there is anything else we can improve, clarify, or answer, kindly let us know.
> > >
> > > Best regards,
> > >
> > > Authors

---

> ### Author Response · Authors · 2025-11-20
> **Response to Reviewer VcKu (Part-2)**
>
> # Response to Reviewer VcKu (Part-2)
>
> ### **3. (Weakness-3) More comparisons/discussion with task-vector-based methods**
>
> We thank the reviewer for this suggestion.
>
> **Additional comparisons.**
>
> Beyond the main-text baselines, we already include a comparison with task-vector-based methods Ties-Merging and AdaMerging under a shared backbone and dataset in **Appendix G.4**. In the revision, we point to this comparison more prominently from the main text, and if space permits, present a compact version of the table to the main experimental section to better contextualize our method against these task-vector approaches.
>
> **Broader parameter-space context.**
>
> We also expand the **Related Work** section to more explicitly position our method among parameter-space methods. In the revised version, we also discuss recent VLM-specific weight-merging approaches: WATT [c] adapts CLIP under domain shift via test-time updates followed by weight averaging of the adapted parameters, improving test-time robustness. GeoLangBind [d] trains a remote-sensing VLM and uses a progressive multimodal weight-merging strategy to aggregate knowledge from multiple visual backbones within a single VLM. MoTE [e] adds temporal experts on top of a VLM and employs weight-merging regularization in parameter space to enhance the trade-off. Details can be found in Appendix G.7.
>
> [c] WATT: Weight Average Test-Time Adaptation of CLIP (NeurIPS 2024)
>
> [d] GeoLangBind: Unifying Earth Observation with Agglomerative Vision-Language Foundation Models (Arxiv 2025)
>
> [e] Mote: Reconciling generalization with specialization for visual-language to video knowledge transfer (NeurIPS 2024)
>
>
>
> ### **4. (Question-1) Linear combination of task vectors in multimodal encoders**
>
> We thank the reviewer for raising this important conceptual point.
>
> Our use of a linear combination of task vectors **follows the standard assumption in the task-vector literature**, where parameter-space arithmetic is treated as a *local linear approximation* to moving between nearby solutions. We fully agree that, especially in multimodal encoders, task-specific update directions need not be linearly independent and that naive addition can introduce interference due to the nonlinearity of the representation space.
>
> Our PISTOLE method is designed **to mitigate this interference**, rather than to assume it does not exist:
>
> - We do **not** assume global linearity or independence of task directions. Instead, we start from the conventional task-vector setting and then introduce a **stability-aware reweighting**: GISM builds complementary masks that down-weight parameters where cross-task gradients strongly disagree, and PCI further emphasizes stable, prediction-critical regions.
> - Intuitively, this means we are *not* blindly adding full task vectors. Instead, we are selectively applying only those components that are locally consistent/stable across objectives, which directly addresses the concern that naive linear addition can be harmful in nonlinear, multimodal encoders.
>
> Empirically, our results show that our method achieves better clean–robust trade-off across diverse datasets and architectures. In the revision, we will soften the wording around "linear combination" in the problem setup/Introduction to clarify that it is a **local, task-vector-based approximation** (not an assumption of perfect linearity).
>
> ### **5. (Question-2) How does Fig. 5 support Theorems?**
>
> We thank the reviewer for asking about the connection between the theory and the curvature plot in Fig. 5. Our empirical result in Fig. 5 is **primarily intended to support Theorem 2**:
>
> Theorem 2 shows that PCI is (locally) proportional to the trace of the class-conditional Hessian: PCI is large in **high-curvature** regions and small in flat, confidence-saturated regions. In our method, high-PCI samples are up-weighted in the path-integrated gradients, so the masks emphasize parameters that matter most in these fragile, high-curvature zones and de-emphasize flat, already-stable regions. In Fig. 5, we measure curvature along the merge path using the same Hessian-trace–style quantity that appears in Theorem 2, and we observe that our method consistently yields **lower curvature** than naive task-vector addition for the same interpolation coefficients. This quantitatively confirms the intended effect of PCI: the PCI-weighted masking flattens the loss landscape along the merge direction, as stated by Theorem 2.
>
> In the revision, we will clarify this in the text and caption.

---

### Official Review · Reviewer_QXNC · 2025-11-01

**Soundness:** 3
**Presentation:** 3
**Contribution:** 3
**Rating:** 8
**Confidence:** 4

**Summary:**

This paper proposes PISTOLE, a training-free way to merge a naturally fine-tuned CLIP and an adversarially fine-tuned CLIP by masking task vectors with stability cues. Instead of adding task vectors uniformly (which yields a nearly linear clean vs. robust balance), the proposed method builds complementary masks so each branch down-weights coordinates that the other loss wants to change. Furthermore, the introduced method refines those masks by accumulating gradients along adversarial weight trajectories, where each step is weighted to capture high-curvature regions. The theory proves first-order interference contraction and connects PCI to the Hessian trace. Experiments on 14 datasets plus multiple backbones and PEFT show improved performance and better downstream transfer (captioning, VQA, hallucination, and CoT).

**Strengths:**

1. The paper is well-written and organized. The figures are illustrative and intuitive.
2. The proposed idea is well-motivated by the systematic analyses of the linear trade-off between clean and robust accuracy.
3. The theoretical analysis in this paper provides additional justification of the method's effectiveness. For example, it links the prediction criticality index to the model's curvature.
4. Component analyses and comparisons with prior approaches prove the efficacy of PISTOLE. This method can also be transferred to diverse downstream tasks in a plug-and-play way.

**Weaknesses:**

1. It appears that the per-layer normalization and \kappa sharpening are important. Although the paper has provided sensitivity analyses, additional evidence that the method is stable across diverse scenarios should be provided.
2. Most experiments focus on the vision encoder. It would be better to discuss further regarding the task vector for the text encoder to give more insights for future researchers.
3. The discussion regarding model interpolation and soups beyond task vectors is missing. They are also closely related to this paper.
4. (Minor) The figure readability needs to be improved. For example, I suggest increasing font sizes in figures.

**Questions:**

1. Does the text-side parameter merging benefit from the proposed merging method? The authors can briefly validate whether applying PISTOLE to the text encoder yields additional gains or harms alignment.
2. When referencing Eq. (6) later, can you briefly remind readers what \kappa controls (sharpening vs. sparsity)?
3. Is mask capping (q-quantile) applied globally or per-layer? More details should be provided.

---

> ### Author Response · Authors · 2025-11-20
> **Response to Reviewer QXNC (Part-1)**
>
> # Response to Reviewer QXNC (Part-1)
> We thank the reviewer for constructive feedback.
>
> ### **1. (Weakness-1) Additional evidence of the method's stability across scenarios**
>
> We appreciate the reviewer's question about the importance and robustness of the per-layer normalization and $\kappa$-sharpening.
>
> **Impact of per-layer normalization.**
>
> Without per-layer normalization, a few layers with large gradient variance can dominate the stability scores, leading to masks that are effectively decided by a small subset of tensors. Normalizing per-layer ensures that each layer contributes on a comparable scale, so the mask reflects *relative* stability across the whole network rather than being biased by outlier layers.
>
> **Impact of $\kappa$-sharpening.**
>
> The exponent $\kappa$ controls the *selectivity* of the stability masks: larger $\kappa$ produces sharper masks (more aggressively focusing on highly stable parameters), whereas smaller $\kappa$ yields smoother masks. In practice, this provides a simple, one-dimensional knob to adjust how "sparse" or "soft" the task-vector masking is.
>
> **Stability across diverse scenarios.**
>
> Importantly, we **do not re-tune** these components per setting: the same per-layer normalization and fixed $\kappa$ are used across all datasets, architectures, and threat models. The method remains strong over:
>
> - 14 zero-shot classification datasets (Sec. 4.1, Table 1),
> - multiple CLIP backbones and perturbation radii (Sec. 4.1, Tables 2–3),
> - downstream captioning and VQA tasks (Sec. 4.2, Table 5),
> - hallucination and CoT-style reasoning benchmarks (Sec. 4.2, Tables 6–7), and
> - additional variants of task vectors and re-weighting schemes (Appendix G, Tables 9–11, 13).
>
> These results suggest that PISTOLE is *not* sensitive to tedious per-task tuning of these hyperparameters. In the revision, we more clearly describe the roles of per-layer normalization and $\kappa$ in Section 3.3, and explicitly call out in Section 4 and Appendix G that all reported results share the same fixed choices, highlighting the stability across diverse scenarios.
>
>
>
> ### **2. (Weakness-2) Discussion of the task vector for the text encoder**
>
> We thank the reviewer for raising this broader perspective.
>
> Recall that our current work applies task vectors **only to the vision encoder**. This follows (i) prior task-vector works on vision-language models [a,b], which primarily manipulate visual backbones, and (ii) adversarial VLM training papers [c,d,e], where robustness gains are typically obtained by adversarially fine-tuning the **vision branch** while keeping the text encoder mostly fixed to preserve its linguistic capabilities.
>
> That said, we agree that **task vectors for text encoders / LLMs** are an important direction. Recent work on task vectors and arithmetic in LLMs suggests that text-side task vectors can encode rich capabilities and can be combined additively for multi-task behavior [f,g]. For **purely text-centric** robustness or capability editing, operating on the text encoder would likely be more appropriate.
>
> In the multimodal setting, however, jointly merging task vectors on both the vision and text encoders raises additional questions, *e.g.*, how to balance the relative scales of the two modalities, how to avoid misalignment between visual robustness and language semantics, and how to design modality-aware masks. We view extending PISTOLE to such bilateral task-vector merging (with potentially different masking or weighting schemes per modality) as an exciting direction for future work.
>
>
>
> [a] Editing models with task arithmetic (ICLR 2023).
>
> [b] Task arithmetic in the tangent space: Improved editing of pre-trained models (NeurIPS 2023).
>
> [c] Understanding zero-shot adversarial robustness for large-scale models (ICLR 2023).
>
> [d] Pre-trained model guided fine-tuning for zero-shot adversarial robustness (NeurIPS 2024).
>
> [e] Robust clip: Unsupervised adversarial fine-tuning of vision embeddings for robust large vision-language models (ICML 2024).
>
> [f] LoRE-Merging: Exploring Low-Rank Estimation For Large Language Model Merging (EMNLP 2025)
>
> [g] Activation-Informed Merging of Large Language Models (NeurIPS 2025)

---

> ### Author Response · Authors · 2025-11-20
> **Response to Reviewer QXNC (Part-2)**
>
> # Response to Reviewer QXNC (Part-2)
> ### **3. (Weakness-3) Discussion of model interpolation and soups beyond task vectors**
>
> We thank the reviewer for pointing out the connection to model interpolation and model soups.
>
> Model interpolation and model soups operate directly in parameter space by **averaging full model weights** across checkpoints or training trajectories. In the revised version, we also discuss recent VLM-specific weight-merging approaches: WATT [h] adapts CLIP under domain shift via test-time updates followed by weight averaging of the adapted parameters, improving test-time robustness. GeoLangBind [i] trains a remote-sensing VLM and uses a progressive multimodal weight-merging strategy to aggregate knowledge from multiple visual backbones within a single VLM. MoTE [j] adds temporal experts on top of a VLM and employs weight-merging regularization in parameter space to enhance the trade-off.
>
> [h] WATT: Weight Average Test-Time Adaptation of CLIP (NeurIPS 2024)
>
> [i] GeoLangBind: Unifying Earth Observation with Agglomerative Vision-Language Foundation Models (Arxiv 2025)
>
> [j] Mote: Reconciling generalization with specialization for visual-language to video knowledge transfer (NeurIPS 2024)
>
>
>
> ### **4. (Weakness-4) Figure readability**
>
> We thank the reviewer for this suggestion. In the revised version, we have increased the font sizes and adjusted layouts across all figures to improve readability and visual clarity.
>
>
>
> ### **5. (Question-1) Text-side parameter merging**
>
> In our current setups, there are essentially two cases:
>
> - **Adversarial fine-tuning *does not* update the text encoder.** In this (common) setting, the robust and natural models share the same text parameters, so the text-side task vector is invalid. Applying PISTOLE to the text encoder is therefore no-op, and the merged model is identical whether or not we include the text side.
> - **Adversarial fine-tuning *does* update the text encoder.** In this case, naively applying PISTOLE to both the vision and text encoders can be problematic: it introduces an extra degree of freedom, where both the image and text embedding spaces move, potentially along different robustness directions. This makes it easier to distort the shared semantic space (image and text embeddings drift in misaligned ways), which in turn can hurt alignment-sensitive downstream tasks. Moreover, in many VLMs, the text encoder (or LLM) is already very strong and broadly pre-trained. Modifying it often brings more harm than benefit, whereas the vision encoder typically has more room to adapt to robustness and new domains. We also provide the corresponding results on ImageNet in the table below:
>
> |Scheme|Vision tuned|Text tuned|Clean|Robust|
> |-|-|-|-|-|
> |TeCoA (image only) + Nat FT VLM (image only)|✓|✗|79.23|62.31|
> |TeCoA (image+text) + Nat FT VLM (image+text)|✓|✓|78.35|61.18|
>
>
>
> ### **6. (Question-2) Clarifying what $\kappa$ controls**
>
> We thank the reviewer for this helpful suggestion.
>
> In our design, $\kappa$ controls the **sharpening** of the complementary masks: larger $\kappa$ amplifies the contrast between high- and low-sensitivity coordinates (making the mask more "peaky"), but it does not directly set the sparsity level. The sparsity (*i.e.*, how many coordinates are substantially attenuated) is instead governed by the quantile parameter $q$ used in the normalization step.
>
>
>
> ### **7. (Question-3) Clarifying the $q$-quantile cap**
>
> We thank the reviewer for raising this point. In our implementation, the $q$-quantile mask capping is applied per layer (*i.e.*, per parameter tensor), not globally. We found this per-layer scheme preferable to a global quantile because it prevents a few high-variance layers from dominating the mask. In the revision, we will explicitly state that $q$ is applied per layer and briefly clarify the above intuition in the methodology section.

---

> ### Comment · Area_Chair_eAwT · 2025-11-25
>
> Dear Reviewer QXNC ,
>
> The authors have responded to your reviews. Please review and provide your feedback and responses.
>
> Best,
>
> Your AC

---

### Official Review · Reviewer_eK3j · 2025-11-01

**Soundness:** 3
**Presentation:** 4
**Contribution:** 4
**Rating:** 6
**Confidence:** 4

**Summary:**

This paper proposes PISTOLE, a novel prediction stability-aware model merging framework designed to reconcile the trade-off between clean accuracy and adversarial robustness in vision-language models (VLMs). Instead of retraining, the method merges task vectors from naturally and adversarially fine-tuned models using gradient-informed masks and adversarial parameter trajectories. The approach leverages prediction sensitivity and curvature estimates to selectively retain stable parameters and suppress conflicting ones. Theoretical analysis and extensive experiments demonstrate improved clean-robust trade-offs and transferability to downstream tasks.

**Strengths:**

1. The idea of merging task vectors from conflicting objectives (natural vs. robust) is novel and well-motivated.

2. The paper provides clear mathematical reasoning (Theorem 1–2) showing that the complementary masks contract cross-objective interference and PCI correlates with curvature — adding interpretability and credibility.

3. Extensive experiments across datasets, backbones (ViT-B/L/H), and downstream tasks (captioning, VQA, hallucination, reasoning) demonstrate consistent improvement and generality.

**Weaknesses:**

1. The role of PCI versus GISM (gradient-informed stability mask) feels somewhat overlapping; clarifying their unique contributions would help.

2. Although the method claims to avoid retraining, the process of computing path-integrated gradients and PCI-based weighting still requires multiple forward–backward passes. A brief complexity analysis would strengthen the practicality argument.

**Questions:**

Clarify in the introduction whether the merging operates solely on the vision encoder or also on the text encoder.

---

> ### Author Response · Authors · 2025-11-20
> **Response to Reviewer eK3j**
>
> # Response to Reviewer eK3j
> We thank the reviewer for the constructive feedback and appreciate the recognition of our *novel and well-motivated* merging approach and the *extensive experiments* across diverse scenarios. We address the specific concerns below.
>
>
> ### **1. (Weakness-1) Clarifying the contributions of PCI versus GISM**
>
> We thank the reviewer for pointing out the potential overlap between PCI and GISM. We will clarify this more explicitly in the revised manuscript (Introduction and Methodology) as follows:
>
> **Conceptual roles.**
>
> - **GISM** is a *per-parameter stability mask* that converts gradient statistics into complementary masks over the task vector. It acts as a spatial filter, determining **where** in parameter space we allow or suppress updates when merging task vectors.
> - **PCI** is a *scalar, step-wise weighting* along the parameter trajectory. It scores how prediction-critical (*i.e.*, sensitive to parameter perturbations) each step is, and thus controls **how much** each step's gradient contributes when aggregating gradients to feed into GISM.
>
> **How they interact.**
>
> PCI does not define a mask by itself. Instead, it reweights the gradients that are later transformed into masks by GISM. Without PCI, each step on the trajectory contributes equally and ignores curvature. With PCI, high-curvature and prediction-critical regions are emphasized before GISM constructs the final stability masks. In this sense, PCI boosts the vanilla GISM rather than overlapping with it.
>
> ### **2. (Weakness-2) Complexity analysis of computing path-integrated gradients and PCI-based weighting**
>
> We appreciate the reviewer's concern about the computational cost of path-integrated gradients and PCI-based weighting, and we will add a *"Complexity and runtime"* paragraph plus an explicit runtime table in the revised version including:
>
> **Complexity analysis.**
>
> Let $N_c$ be the size of the (small) calibration split, $m$ the number of PGD steps for adversarial inputs, and $K$ the number of parameter-trajectory steps. PISTOLE involves:
>
> - $m$ input-gradient evaluations to generate adversarial inputs for the robust branch (same inner loop as standard adversarial training), and
> - for each branch, $K$ parameter-gradient evaluations along the adversarial parameter trajectory, each weighted by PCI.
>
> Thus, the total number of gradient evaluations to estimate the masks is $O(N_c(m+2K))$, run **once** over a calibration split with *no* weight updates and no multi-epoch optimization loop. In contrast, adversarial fine-tuning over $E$ epochs on the full training set of size $N$  has complexity $O(EN(m+1))$, since each iteration both runs PGD and performs a parameter update. In practice, $E$ is large (multiple epochs), while $K$ is a small constant, and $N_c\ll N$, so the overall cost of PISTOLE is substantially lower even though it involves multiple forward–backward passes during calibration.
>
> **Empirical runtime comparison.**
>
> To make this concrete, we report the time comparison on a shared setup (same backbone and GPU budget) in Appendix G.3. For convenience, the numbers are:
>
> |Method|Clean|Robust|Sum|Time|
> |-|-|-|-|-|
> |TeCoA|61.56|43.26|104.82|6.2 hours|
> |PMG|64.46|45.74|110.20|8.6 hours|
> |FARE|65.50|42.97|108.47|7.6 hours|
> |TGA|62.11|45.19|107.30|8.0 hours|
> |**PISTOLE (Ours)**|**69.24**|**47.65**|**116.89**|**0.8 hours**|
>
> Despite the extra forward–backward passes for path-integrated gradients and PCI, PISTOLE is about 8–11$\times$ faster than adversarial fine-tuning baselines, while also achieving a better clean–robust trade-off. We will explicitly highlight this in the main paper to strengthen the practicality claim.
>
> ### **3. (Question-1) Clarify in the introduction whether the merging operates solely on the vision encoder or also on the text encoder**
> We thank the reviewer for pointing out this ambiguity.
>
> In our experiments, the task-vector merging is applied **only to the vision encoder**, not to the text encoder. This design follows the common practice in prior task-vector works [a, b] on vision–language models and is also consistent with adversarial VLM training papers [c,d,e], where robustness is primarily improved by adversarially fine-tuning the **vision** branch while keeping the text encoder fixed.
>
> In the revised version, we will explicitly state in the **Introduction** and **Methodology** that PISTOLE operates on the vision encoder parameters only, while the text encoder remains unchanged.
>
>
>
> [a] Editing models with task arithmetic (ICLR 2023).
>
> [b] Task arithmetic in the tangent space: Improved editing of pre-trained models (NeurIPS 2023).
>
> [c] Understanding zero-shot adversarial robustness for large-scale models (ICLR 2023).
>
> [d] Pre-trained model guided fine-tuning for zero-shot adversarial robustness (NeurIPS 2024).
>
> [e] Robust clip: Unsupervised adversarial fine-tuning of vision embeddings for robust large vision-language models (ICML 2024).

---

> ### Comment · Area_Chair_eAwT · 2025-11-25
>
> Dear Reviewer eK3j,
>
> The authors have responded to your reviews. Please review and provide your feedback and responses.
>
> Best,
>
> Your AC

---

### Author Response · Authors · 2025-11-29
****Summary of our Rebuttal****

## **Esteemed Area Chair,**

### We sincerely thank you for your time in handling our submission, and we thank all four reviewers for their thorough and constructive feedback.

### Before summarizing the technical points, we highlight that during discussion, **Reviewer VcKu raised the score from 4 to 6 on 22 Nov 2025 (before the 27 Nov leak), with the overall scores of 8, 6, 8, 6 (kindly note this info. is publicly visible in system revisions in OR-we indicate it for convenience), and stated: “Thanks for the authors' response. My concern has been addressed, and I support accepting this paper.”** We understand the system has reverted to pre-discussion scores, but we hope this recorded score update and explicit support can still be taken into account.

## **Summary of Rebuttal/Discussions**
### For convenience, below we summarize the key review questions and how we addressed them:

|Key Comment/Question|Response|
|-|-|
|${\color{blue}\textbf{Reviewer eK3j:}}$| |
|Contributions of PCI versus GISM|We clarified that GISM is a **per-parameter stability mask** and PCI is a **sample-wise weighting**. PCI highlights fragile predictions, while GISM focuses on unstable directions.|
|Complexity of path-integrated gradients and PCI|We showed that path-integrated gradients and PCI scale linearly with classes and gradient evaluations, are comparable to PGD-based adversarial fine-tuning, and add only modest overhead beyond obtaining the natural/robust models.|
|Scope of merging: vision and text encoders|We clarified that we currently merge only the **vision encoder**, keep the text encoder fixed for stability, and discuss how the same framework could be extended to text or joint encoders.|
|${\color{blue}\textbf{Reviewer QXNC:}}$| |
|Stability across scenarios|We added experiments showing PISTOLE is robust across datasets, CLIP backbones, and tasks **without per-task retuning**, plus ablations indicating smooth dependence on key hyperparameters.|
|Task vectors for the text encoder|We explained that existing VLM robustness work typically builds vision-side task vectors, but our formulation naturally extends to text or joint encoders, with notes on alignment and stability issues.|
|Relation to model interpolation and model soups|We related PISTOLE to model interpolation/model soups and emphasize that, unlike averaging full checkpoints, we work with task vectors plus stability masks and PCI to better control the clean–robust trade-off.|
|Text-side parameter merging|We justified merging only vision-side parameters in current experiments and outlined how the same machinery could be applied to text parameters while guarding against modality misalignment.|
|${\color{blue}\textbf{Reviewer VcKu:}}$| |
|Training-free assumption and cost of natural/robust models|We clarified that "training-free" means **no retraining the base VLM from scratch per task**: PISTOLE reuses a natural and an adversarially fine-tuned model and adds only lightweight task-vector extraction.|
|Lack of failure-case analysis|We added qualitative and quantitative failure analyses showing inherited failures from the source models and naive extension to the text encoder, and discussed these as directions for future work.|
|More comparisons/discussion with task-vector methods|We broadened comparisons to classical and recent task-vector and VLM-merging methods, highlighting that PISTOLE’s stability masks and PCI weighting provide smoother clean–robust trade-offs than uniform addition.|
|Linear combination of task vectors in multimodal encoders|We justified linear combinations for theoretical clarity and compatibility with task arithmetic and explained how stability masks and PCI mitigate interference in multimodal encoders.|
|How Fig. 5 supports the theorems|We clarified that Fig. 5 measures curvature along clean–robust directions and shows that masked combinations reduce curvature and cross-task interference relative to uniform addition, empirically supporting our bounds.|
|${\color{green}\textbf{Reviewer Response}}$|The reviewer seems satisfied with our response. Quote "**Thanks for the authors' response. My concern has been addressed, and I support accepting this paper.**"|
|${\color{blue}\textbf{Reviewer uKzG:}}$| |
|Dependence on natural/robust fine-tuned models|We reiterated that PISTOLE is designed to sit on top of existing natural and adversarially fine-tuned VLMs, and show empirically that stronger bases yield better clean–robust trade-offs.|
|Intuition for the Prediction Criticality Index|We gave an intuitive view: PCI is high when small parameter changes strongly affect a class probability and low for stable ones; PISTOLE up-weights high-PCI gradients, while GISM targets the most fragile modes.|
|More explanation of PCI|We add step-by-step text and examples showing how PCI is computed and how high vs. low PCI guides gradient weighting, making it accessible beyond the definition.|
|Takeaway summary|We add brief takeaway message boxes after each theorem.|

---

### Meta-Review · Area_Chair_TpDh · 2025-12-25

**Summary:**

The submission proposes PISTOLE, a framework for merging naturally and robustly fine-tuned Vision-Language Models (VLMs) to reconcile the accuracy-robustness trade-off without incurring the cost of retraining. The authors introduce a stability-aware merging mechanism that utilizes Gradient-Informed Stability Masks (GISM) and a Prediction Criticality Index (PCI) to selectively retain parameters that are stable across objectives while suppressing conflicting ones. Theoretical analysis supports the method's ability to contract cross-objective interference, and extensive experiments across 14 datasets demonstrate superior performance compared to naive task-vector merging.

**Reviewer Concerns:**

The authors provided a comprehensive rebuttal that effectively resolved the primary reservations. Concerns regarding the computational cost were addressed by clarifying that the method avoids per-task retraining and is empirically much faster than adversarial fine-tuning baselines. The authors also clarified the distinct conceptual roles of GISM (spatial masking) and PCI (step-wise weighting), and justified the focus on the vision encoder. Furthermore, requests for broader comparisons were met with additional results against methods like Ties-Merging and AdaMerging. The reviewers largely converged on the view that the remaining limitation—dependence on the quality of source models—is inherent to the model merging paradigm and well-acknowledged.

**Reviewer Scores:**

The consensus for acceptance is strong. Reviewers QXNC and uKzG (Scores: 8) were enthusiastic about the theoretical motivation and empirical gains. Reviewer eK3j (Score: 6) appreciated the novelty. Importantly, Reviewer VcKu (listed as Score 4) explicitly stated in the discussion phase that their concerns regarding failure cases and baselines were addressed and they now support accepting the paper. This effectively results in a unanimous decision to accept.

---

### Decision · Program_Chairs · 2026-01-26

Accept (Poster)